# Copper bioavailability is a KRAS-specific vulnerability in colorectal cancer

Léo Aubert[1,11], Neethi Nandagopal[1,11], Zachary Steinhart[2], Geneviève Lavoie[1], Sami Nourreddine[1], Jacob Berman[3], Marc K. Saba-El-Leil[1], David Papadopoli[4], Sichun Lin[2], Traver Hart[5], Graham Macleod[2], Ivan Topisirovic[4], Louis Gaboury[1,6], Christoph J. Fahrni[7], Daniel Schramek[3,8], Sylvain Meloche[1,9], Stephane Angers[2,10] & Philippe P. Roux[1,6✉]

Despite its importance in human cancers, including colorectal cancers (CRC), oncogenic KRAS has been extremely challenging to target therapeutically. To identify potential vulnerabilities in KRAS-mutated CRC, we characterize the impact of oncogenic KRAS on the cell surface of intestinal epithelial cells. Here we show that oncogenic KRAS alters the expression of a myriad of cell-surface proteins implicated in diverse biological functions, and identify many potential surface-accessible therapeutic targets. Cell surface-based loss-of-function screens reveal that ATP7A, a copper-exporter upregulated by mutant KRAS, is essential for neoplastic growth. ATP7A is upregulated at the surface of KRAS-mutated CRC, and protects cells from excess copper-ion toxicity. We find that KRAS-mutated cells acquire copper via a non-canonical mechanism involving macropinocytosis, which appears to be required to support their growth. Together, these results indicate that copper bioavailability is a KRAS-selective vulnerability that could be exploited for the treatment of KRAS-mutated neoplasms.

[1] Institute for Research in Immunology and Cancer (IRIC), Université de Montréal, Montreal, 2950, Chemin de la Polytechnique, Montréal, QC H3T 1J4, Canada. [2] Department of Pharmaceutical Sciences, Leslie Dan Faculty of Pharmacy, University of Toronto, 144 College Street, Toronto, ON M5S 3M2, Canada. [3] Lunenfeld-Tanenbaum Research Institute, Mount Sinai Hospital, University of Toronto, 600 University Ave., Toronto, ON M5G 1X5, Canada. [4] Department of Oncology, Lady Davis Institute for Medical Research, Jewish General Hospital, McGill University, 3755 Côte Sainte-Catherine Road, Montreal, QC H3T 1E2, Canada. [5] MD Anderson Cancer Center, University of Texas, 515 Holcombe Blvd, Houston, TX 77030, USA. [6] Department of Pathology and Cell Biology, Faculty of Medicine, Université de Montréal, Montreal, QC, Canada. [7] School of Chemistry and Biochemistry, Petit Institute for Bioengineering and Bioscience, Georgia Institute of Technology, 315 Ferst Dr NW, Atlanta, GA 30332, USA. [8] Department of Molecular Genetics, University of Toronto, Toronto, ON, Canada. [9] Department of Pharmacology and Physiology, Faculty of Medicine, Université de Montréal, Montreal, QC, Canada. [10] Department of Biochemistry, Faculty of Medicine, University of Toronto, Toronto, ON, Canada. [11] These authors contributed equally: Léo Aubert, Neethi Nandagopal. ✉email: philippe.roux@umontreal.ca

Colorectal cancer (CRC) is one of the leading causes of cancer-related deaths worldwide[1]. CRC is a highly heterogeneous disease, composed of several biologically and clinically distinct entities that can determine the prognosis and response to therapy. Identification of subtype-specific vulnerabilities is necessary for the design of tailored treatment and improved patient outcomes[2]. In the metastatic setting, treatment options for CRC are limited, but recent advances in targeted therapies, such as antiVEGF-A and antiEGFR antibodies, have improved overall survival[3,4]. Unfortunately, nearly half of CRC cases harbor KRAS mutations, which are negative predictive markers for antiEGFR therapy[5] and common drivers of innate and acquired resistance[6] (Supplementary Fig. 1a).

KRAS belongs to the RAS family of ubiquitously expressed small GTPases, which serve as a major communication hub between cell-surface receptors and intracellular signaling pathways[7]. Cancer-causing KRAS mutations result in aberrant activation of the RAS/mitogen-activated protein kinase (MAPK) pathway[8,9]. This provides diverse advantages to cancer cells, such as increased proliferation and survival, macropinocytosis and altered metabolism, evasion of the immune response, alteration of the tumor microenvironment, and metastasis[9]. While several decades of intense efforts have been unsuccessful to pharmacologically target oncogenic forms of KRAS[10], the recent emergence of clinical-grade KRAS$^{G12C}$ inhibitors has shown promise in the management of tumors harboring oncogenic KRAS$^{G12C}$ mutations[11]. However, recent reports have highlighted the partial effects of these inhibitors in patients and uncovered many mechanisms of resistance that involves compensatory pathways, such as EGFR, FGFR, AXL, and PI3K/AKT[12,13]. Therefore, the development of alternative therapeutic options is still urgently needed for the efficient targeting of KRAS-addicted cancers[10].

While the intracellular signaling events modulated by KRAS are currently being evaluated as potential therapeutic targets[14,15], much less is known about its potential impact on the cell surface. Elucidating how oncogenic KRAS modifies the cell surface proteome (surfaceome) may improve our understanding of its complex mechanism of action, and possibly identify new attractive therapeutic targets. To achieve this, we combine both cell-surface proteomics and loss of function CRISPR screens to identify a novel candidate, the copper (Cu)-exporter ATP7A as synthetic lethal to KRAS-transformed cells. ATP7A regulates the intracellular Cu levels and tumor growth in KRAS-transformed cells. This study demonstrates that inhibiting Cu supply is an attractive therapeutic option for KRAS-mutant cells.

## Results

**KRAS$^{G12V}$ reprograms the intestinal cell surfaceome**. To identify surfaceome changes associated with KRAS-mediated transformation in the context of CRC development, we used an isogenic intestinal epithelial cell model (IEC-6) stably expressing KRAS$^{G12V}$ (KRAS) or an empty vector (Control). IEC-6 cells normally form a confluent cobblestone-like monolayer (Supplementary Fig. 1b), but as expected, ectopic expression of KRAS$^{G12V}$ reduced cell–cell contact inhibition and confers a proliferative advantage under both adherent (Supplementary Fig. 1c) and nonadherent (Supplementary Fig. 1d) conditions. To determine the impact of oncogenic KRAS on the intestinal epithelial cell transcriptome, we performed genome-wide RNA-sequencing (RNA-seq) and identified a large number of upregulated (546) and downregulated (1225) genes (Fig. 1a, Supplementary Data 1). Using gene set enrichment analysis (GSEA)[16], we found a significant overlap with hallmark signatures associated with KRAS signaling (Fig. 1b) and MYC targets (Supplementary Fig. 1e), which are well-known KRAS downstream

effectors. In addition, we found a very good correlation between our dataset and oncogenic KRAS-specific transcriptional changes occurring in CRC specimens from The Cancer Genome Atlas (TCGA) (Fig. 1c), which illustrates that the IEC-6 cell model mimics some of the validated transcriptional signature of KRAS-mutated CRC. We also performed a gene-annotation enrichment analysis using g:Profiler[17,18], and found highly significant enrichments in several gene-annotations associated with the cell surface, including the terms plasma membrane and cell periphery (Fig. 1d, e). These results suggested that oncogenic KRAS regulates a transcriptional program that profoundly modifies the intestinal epithelial cell surfaceome.

To characterize changes occurring at the cell surface in response to oncogenic KRAS, we conducted a comprehensive surfaceome analysis based on the coupling of two state-of-the-art chemoproteomic approaches, cell surface biotinylation (CSB) and cell surface capture (CSC) (Supplementary Fig. 2a), with label-free quantification by liquid chromatography and tandem mass spectrometry (LC-MS/MS) (Fig. 1f). While CSB involves the labeling of free primary amine groups in cell-surface proteins with the non-permeable reagent Sulfo-NHS-LC-biotin, CSC is based on the specific labeling of cell-surface sialylated glycoproteins using aminooxy-biotin, taking advantage of the fact that most cell-surface proteins are glycosylated. We confirmed that both surface biotinylation methods resulted in specific cell-surface labeling in IEC-6 cells, with very little evidence of intracellular biotinylation, as shown using immunofluorescence microscopy (Supplementary Fig. 2b). We found that, irrespective of the chemoproteomic method used, KRAS$^{G12V}$ expression drastically modified the cell-surface biotinylation profile of IEC-6 cells (Fig. 1g). Interestingly, we also noticed that the biotinylation pattern was very different depending on the approach used (Fig. 1g), suggesting that using both approaches in parallel may increase the coverage and yield of cell-surface proteins. After data curation to determine high-confidence cell-surface proteins (see Methods section), we identified 366 and 354 highly relevant cell-surface proteins with CSB and CSC, respectively, from which 329 were common to both datasets (Supplementary Fig. 2c). Consistent with the observed differential biotinylation patterns (Fig. 1g), we found that many proteins were exclusively found with either CSB (37) or CSC (25), indicating the complementarity of these two approaches for cell-surface proteomics. Importantly, we found a high level of reproducibility between biological replicates (Supplementary Fig. 2d), and observed a good correlation between both methods ($R^2 = 0.571$) (Supplementary Fig. 2e). Results were combined to find that 60 (~15%) and 85 (~22%) cell-surface proteins were significantly upregulated and downregulated (P < 0.05, unpaired one-tailed Student's $t$-test) in KRAS$^{G12V}$-transformed cells, respectively (Fig. 1h, Supplementary Fig. 2f, Supplementary Data 2).

To determine the involvement of transcriptional regulation in cell-surface reprogramming by KRAS$^{G12V}$, we compared both surfaceome and transcriptome datasets. We found a relatively good correlation ($R^2 = 0.531$) between both datasets (Supplementary Fig. 3a), suggesting that many changes occurring at the cell surface result from KRAS$^{G12V}$-dependent transcriptional regulation (Fig. 1i). Nevertheless, we also found a large number of cell-surface proteins whose expression did not correlate with respective transcript levels (Fig. 1i, Supplementary Data 3), suggesting the involvement of post-transcriptional mechanisms downstream of KRAS$^{G12V}$. We used immunofluorescence microscopy, flow cytometry, and immunoblotting to validate these positive (EGFR, E-cadherin, N-cadherin, and CD44) and negative (ITGB1, EphA2, TLR4, TGFBR1, and AXL) correlations (Supplementary Fig. 3b, c). Together, these data demonstrate that oncogenic KRAS reprograms the surfaceome of intestinal

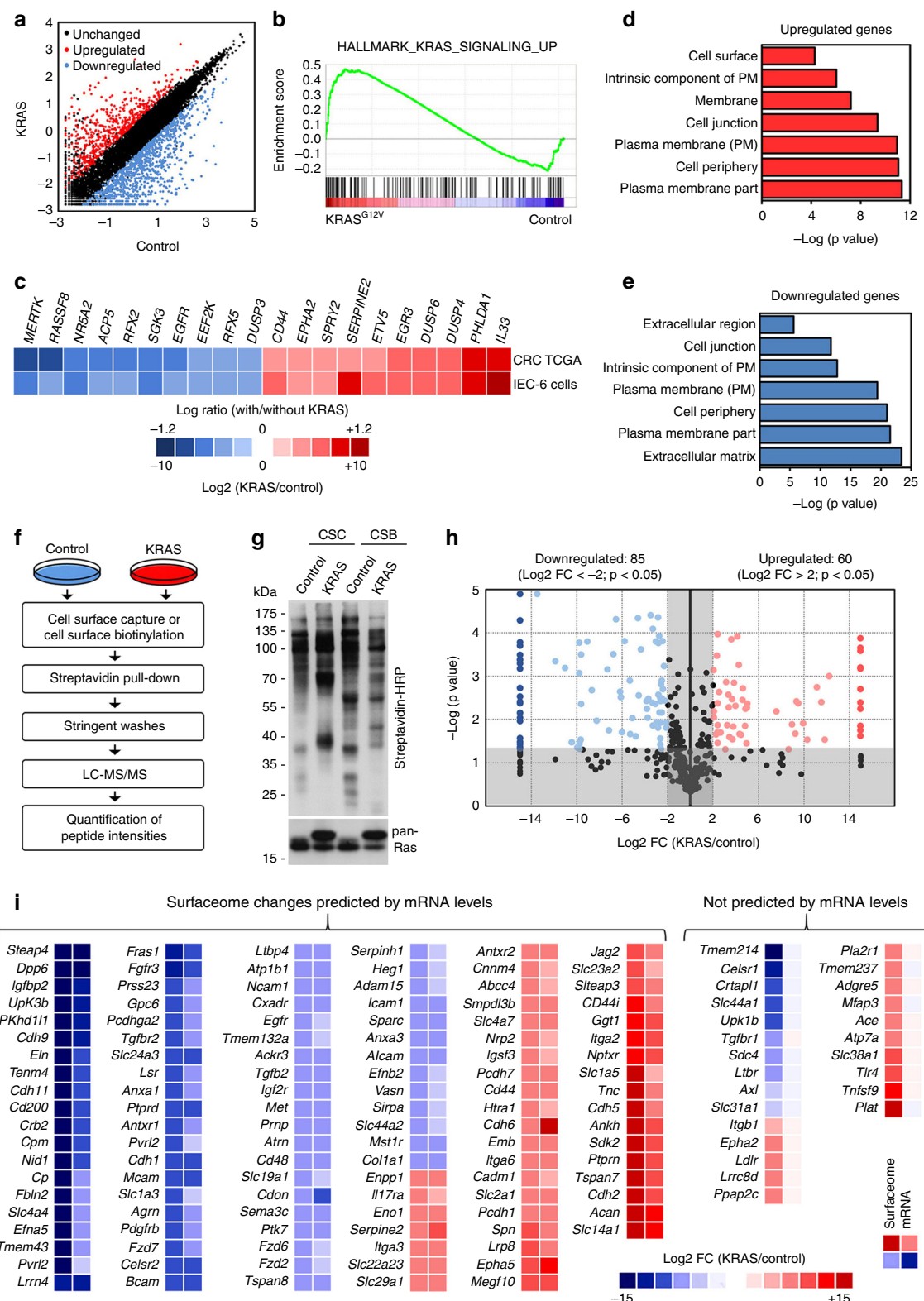

epithelial cells through both transcriptional and post-transcriptional mechanisms.

**ATP7A is a synthetic lethal target for KRAS-addicted CRC.** To identify cell-surface proteins that are required for the fitness of mutant KRAS-mutated cells, we carried out an in vitro loss-of-function CRISPR/Cas9 screen using a lentiviral gRNA library

designed using our surfaceome results (Fig. 2a, Supplementary Data 4). In addition to genes coding for cell-surface proteins (eight gRNAs per gene), our library also included reference gRNAs targeting both essential (including *Myc*) and nonessential (including *Gpr101*) genes[19]. Following KRAS gRNA-library transduction, Control and KRAS cells were allowed to grow for seven (D7) and fourteen (D14) days. Relative gRNA abundance

**Fig. 1 Oncogenic KRAS modifies the surfaceome of intestinal cells. a–e** IEC-6 cells stably expressing KRAS[G12V] (KRAS) or empty vector (Control) were subjected to deep mRNA sequencing, and global transcriptome changes associated with KRAS[G12V] are depicted (**a**). **b** Transcriptome data was analyzed using publicly available hallmark database in GSEA for potential enrichment of KRAS_SIGNALING_UP signature in our dataset. **c** Log2 Fold Changes (FC) of genes known to be regulated by mutant KRAS were compared between our transcriptome and the publicly available KRAS-mutated colorectal cancer TCGA datasets. Bar charts depicting the most significant Gene Ontology (GO) terms enrichment correlating with transcripts either **d** upregulated or **e** downregulated by KRAS[G12V]. **f–i** Cell surface capture (CSC) and cell surface biotinylation (CSB) were used in parallel for the purification of cell-surface proteins. **f** Workflow for the identification and quantification of surfaceome changes induced by KRAS[G12V]. **g** The global cell-surface biotinylation pattern was evaluated in control and KRAS cells by immunoblotting (IB) using HRP-conjugated streptavidin. Results represent $n = 3$ independent experiments. **h** Volcano plot of high-confidence surfaceome changes induced by KRAS[G12V]. Shaded areas represent cut-off range of negative log10 (*P*-value) versus log2 FC (KRAS/control). **i** Heatmap comparing the most significant surfaceome changes (first column), with respective transcriptomic changes (second column), induced by KRAS[G12V]. For panels (**a**, **h**, **i**), data are means of $n = 3$ independent biological replicates. Log2 FC (KRAS/Control) above 2 or below −2 (4-fold) were considered as significantly upregulated (red) or downregulated (blue), respectively. *P*-values were determined by unpaired two-tailed Student's *t*-tests. $P < 0.05$.

was determined at D7 and D14 by high-throughput sequencing and compared to the initial cell populations (D0) (Fig. 2a). To increase the robustness of our findings and limit potential bias due to genetic drifts, this screen was performed on two different pairs of isogenic IEC-6 cells generated at different times. Our results show that the fold-change distribution of gRNAs targeting essential genes was significantly shifted compared to those targeting non-essential genes, indicating that the screens performed as designed (Fig. 2b). A log Bayes Factor (BF) was then calculated for each gene using the BAGEL algorithm, which measures the confidence that knockdown or knockout of a specific gene causes a decrease in fitness[20]. To determine potent KRAS-specific vulnerabilities, we next calculated a differential essentiality score for each gene, which is based on the difference between BF scores observed in KRAS cells and those observed in Control cells (see Methods section). Strikingly, amongst all genes originating from the surfaceome dataset, *Atp7a* was the only candidate showing specific requirement in KRAS compared to Control cells (Fig. 2c, Supplementary Data 4, 5). A closer examination of the CRISPR/Cas9 screen revealed that 7 out of 8 of gRNAs targeting *Atp7a* affected KRAS IEC-6 cells, compared to wild-type counterparts (Fig. 2d, Supplementary Fig. 4a). Interestingly, the relative abundance of these gRNAs was similar to those observed for the essential reference gene *Myc* (Fig. 2d, Supplementary Fig. 4b), while none of the gRNAs targeting the nonessential gene *Gpr101* were differentially depleted in Control and KRAS IEC-6 cells (Fig. 2d, Supplementary Fig. 4c). Consistent with this, the BF score of *Atp7a* reached a level similar to that of the essential reference gene *Myc* (Supplementary Fig. 4d), suggesting the discovery of a potent synthetic lethal target for KRAS-mutated cells. We tested the relative cell proliferation by RNA-interference (RNAi) in KRAS IEC-6 cells (Fig. 2e), as well as in two KRAS-mutated CRC cell lines (HCT116 and SW620) (Fig. 2f), compared to wild-type KRAS counterparts, Control IEC-6 and CACO-2, respectively. This is consistent with the differential essentiality of ATP7A in KRAS-mutant cells, as suggested by the CRISPR screens.

To assess the pathophysiological relevance of our findings, we conducted an in vivo loss-of-function CRISPR/Cas9 screen using KRAS cells xenografted into NSG mice (Fig. 2g). Briefly, cells transduced with the KRAS gRNA-library were selected for one day prior to transplantation, and allowed to form tumors for one month postimplantation (D30). After extraction of primary tumors, relative gRNA abundance was determined by high-throughput sequencing and compared to the initial cell population (D0). The screen results revealed four hits with significant BF, including *Atp7a*, *Met*, *Tln1*, and *Evc2* (Fig. 2h, Supplementary Data 6). Remarkably, the BF for *Atp7a* was similar to that of *Myc*, validating our in vitro findings (Supplementary Fig. 4e) and highlighting the essentiality of ATP7A for KRAS-mutated

tumors. As indicated, we also identified *Met*, *Tln1*, and *Evc2*, for which previous evidence of synthetic lethality with KRAS only exist for MET[21]. To validate the involvement of ATP7A in KRAS-dependent tumor growth, we performed soft agar assays using KRAS IEC-6 (Supplementary Fig. 4f) and KRAS-mutated CRC (Supplementary Fig. 4g) cells, and found that shRNA-mediated ATP7A knockdown reduced anchorage-independent growth. Collectively, these results show that ATP7A plays an essential role in the tumorigenesis of KRAS-addicted cells.

**ATP7A protects KRAS-mutant cells from cuproptosis.** Our surfaceome analysis showed that KRAS cells express high levels of ATP7A compared to Control cells, but that its cell-surface expression does not correlate with mRNA levels (Fig. 1i, Supplementary Fig. 5a). We confirmed this observation by immunofluorescence microscopy (Fig. 3a) and immunoblotting (Supplementary Fig. 5b), which also revealed that ATP7A is more present at the cell surface of KRAS relative to Control cells (Fig. 3a, b). ATP7A is a homeostatic copper-exporter that protects cells from Cu toxicity by maintaining adequate intracellular Cu levels[22]. ATP7A levels are post-transcriptionally regulated by Cu levels[23], suggesting that KRAS[G12V] cells may have increased intracellular Cu levels than Control cells. Consistent with this, we found that mRNA levels of four genes involved in intracellular Cu homeostasis (*Mt1*, *Mt2a*, *Loxl2*, and *Loxl3*) are significantly elevated in KRAS compared to Control cells (Supplementary Fig. 5c).

Depending on its kinetic accessibility, cellular Cu can be divided into a canonical, static pool (i.e., Cu bound to enzymes such as cytochrome c oxidase or SOD1) and an exchangeable, labile pool (i.e., Cu bound to chaperones such as CCS)[24]. The latter form of Cu is more bioavailable and can participate in dynamic cell signaling pathways[25]. We monitored protein levels of the copper chaperone for superoxide dismutase (CCS), which is a well-established readout of Cu bioavailability[26,27] and whose expression is inversely correlated to bioavailable Cu levels in cells[28]. We found lower CCS levels in KRAS compared to Control cells (Fig. 3c Supplementary Fig. 5d), suggesting an increase in bioavailable Cu due to KRAS[G12V]. To corroborate these findings, we used the recently developed ratiometric Cu(I)-selective fluorescent probe, crisp-17, as another readout for bioavailable Cu[25]. The probe was first validated using the cell-permeant complex CuGTSM (glyoxalbis(N[4]-methyl-3-thiosemicarbazonato) copper(II)) and a Cu(I)-specific chelator (PSP-2)[29], which confirmed that the probe reversibly responds to changes in cellular Cu levels (Supplementary Fig. 5e, f). Using crisp-17, we found a significantly higher fluorescence ratio in KRAS IEC-6 cells compared to Control cells, upon addition of the thiol-selective oxidant DTDP (2,2′-dithiodipyridine), suggesting an

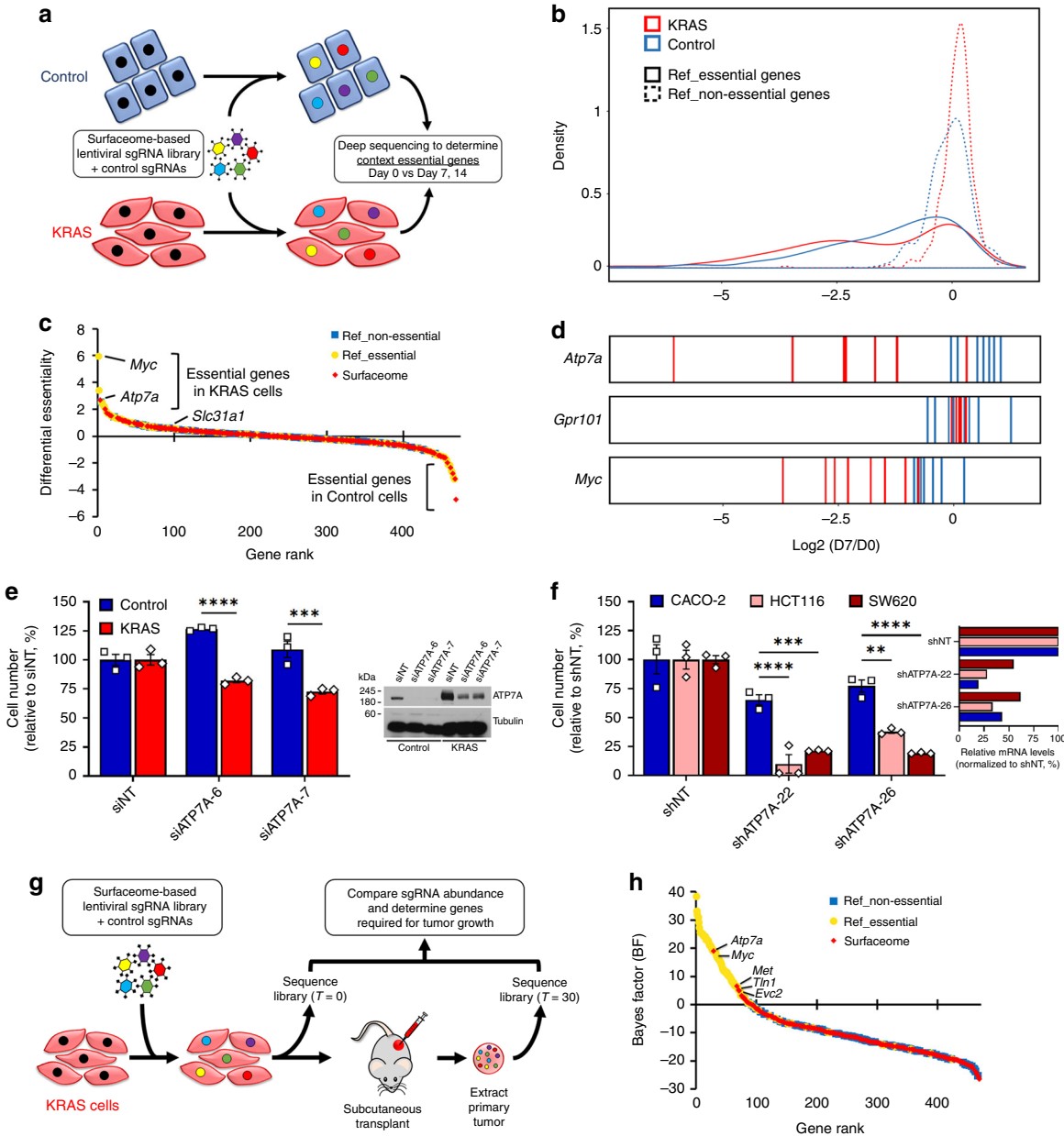

**Fig. 2 Identification of ATP7A as a vulnerability for KRAS-addicted CRC cells. a** Experimental design for identification of KRAS-specific vulnerabilities by in vitro CRISPR/Cas9 screening. **b** Graph depicting the fold-change distributions of gRNAs targeting essential (solid lines) and nonessential (dashed lines) genes at day 7 after infection of Control and KRAS cells with KRAS-library. **c** Waterfall graph showing differential essentiality scores between Control and KRAS cells, depicting genes coding for cell-surface proteins significantly altered by KRAS$^{G12V}$ (red diamonds) and reference genes coding for proteins known to be essential (green circles) or non-essential (blue squares). **d** Bar graphs illustrating the Log2 FC of the eight gRNAs targeting Atp7a, the non-essential gene Gpr101 and essential gene Myc at day 7 in Control (blue) and KRAS (red) cells. **e** Bar chart indicating the relative cell viability of Control and KRAS cells after ATP7A knockdown. IB for ATP7A showing the efficient knockdown by RNAi (right). Data represent $n = 3$ independent experiments with five replicates, and $n = 3$ independent experiments (right). ***$P = 0.0001$; ****$P < 0.0001$. **f** As in **e** CRC cells were analyzed for cell viability upon ATP7A knockdown. Graph showing the efficient knockdown of ATP7A as measured by qPCR (right). Data represent $n = 3$ independent experiments with seven replicates, and two technical replicates (right). ****$P < 0.0001$ (CACO-2 versus HCT116, shATP7A-22); ***$P = 0.0008$ (CACO-2 versus SW620, shATP7A-22); **$P = 0.0027$ (CACO-2 versus HCT116, shATP7A-26); *$P < 0.0001$ (CACO-2 versus SW620, shATP7A-26). **g** Experimental design for the identification of KRAS-specific vulnerabilities in vivo using CRISPR/Cas9. **h** Waterfall graph showing BF scores for genes coding for cell-surface proteins significantly regulated by KRAS$^{G12V}$ (red diamonds), or proteins that are essential (green circles) or non-essential (blue squares). For panels **e**, **f** center values and error bars represent mean ± SEM, and significance was determined using two-way ANOVA with post-hoc Bonferroni's multiple-comparison analysis.

increased pool of labile Cu in KRAS cells (Fig. 3d). Consistent with this interpretation, the kinetics for returning to the basal ratio was slower in KRAS relative to Control cells, suggesting an increased depth of the bioavailable thiol-bound Cu pool in KRAS mutant cells. To determine whether oncogenic KRAS-mediated

ATP7A upregulation provides a protective role against Cu toxicity, we subjected Control and KRAS cells to increasing concentrations of exogenous Cu. Remarkably, we found that Control cells were significantly more sensitive to increasing Cu levels compared to KRAS-mutant cells, and this correlated with

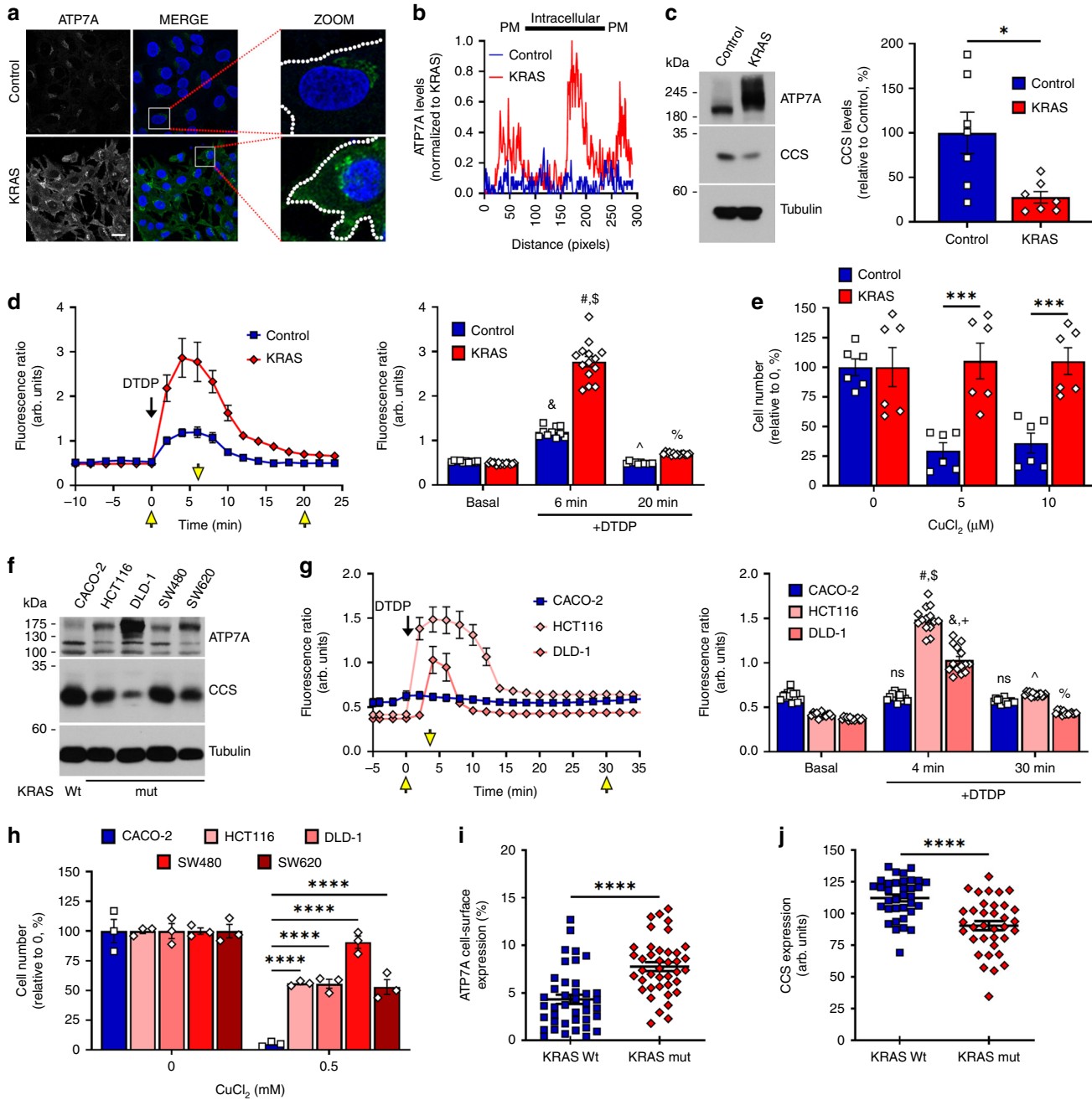

**Fig. 3 ATP7A protects KRAS-mutant cells from cuproptosis. a** Micrographs showing ATP7A expression (green) and DAPI (blue) in Control and KRAS cells. Scale = 20 μm, $n = 3$ independent experiments. **b** Cellular distribution profile of ATP7A as seen in a for Control (blue line) versus KRAS (red line) cells. Data were normalized to KRAS. PM plasma membrane. **c** IB for CCS and ATP7A expression in Control and KRAS cells, with CCS quantifications (right). All panels represent $n = 7$ independent experiments. *$P = 0.0110$. **d** Time-course of average crisp-17 fluorescence ratios ($n = 10$ cells) upon addition of DTDP at $T = 0$ min. Yellow arrows indicate timepoints for which quantifications are shown (right). *P*-values are represented as Control-basal versus DTDP (6 min) (&), Control-DTDP (6 min) versus KRAS-DTDP (6 min) (#), KRAS-basal versus DTDP (6 min) ($), Control-DTDP (6 min) versus DTDP (20 min) (^), KRAS-DTDP (6 min) versus DTDP (20 min) (%). **e** Graph depicting relative cell viability after CuCl2 treatment. Data represent $n = 6$ independent experiments.***$P = 0.0002$ (Control versus KRAS-5); ***$P = 0.0006$ (Control versus KRAS-10). **f** IB depicting ATP7A and CCS levels in the indicated CRC cells. Data represent n = 3 independent experiments. **g** As in **d** but for CRC ($n = 14$ cells). *P*-values are represented as CACO-2-basal versus DTDP (4 min) (ns), CACO-2-DTDP (4 min) versus HCT116-DTDP (4 min) (#), HCT116-basal versus DTDP (4 min) ($), CACO-2-DTDP (4 min) versus DLD-1-DTDP (4 min) (&), DLD-1-basal versus DTDP (4 min) (+), CACO-2-DTDP (4 min) versus DTDP (30 min) (ns), HCT116-DTDP (4 min) versus DTDP (30 min) (^), DLD-1-DTDP (4 min) versus DTDP (30 min) (%). **h** As in **e** but for CRC cells. Data represent $n = 3$ independent experiments.
**i, j** Graphs represent expression of cell-surface ATP7A (**i**, $n = 40$ random images from five patient-derived CRC tumors) and total CCS (**j**, $n = 35$) analyzed by immunohistochemistry. For **c–e**, **g–j** center values and error bars represent mean ± SEM and for left panels (**d**, **g**), mean ± SD. For **d**, **g**, **h** ns not-significant or ****$P < 0.0001$ (&, #, +, $, ^, %). Significance was determined using unpaired two-tailed Student's *t*-tests (**c**, **d**, **g**, **i**, **j**) or two-way ANOVA with post-hoc Bonferroni's multiple-comparison analysis (**e**, **h**).

ATP7A levels (Fig. 3e, Supplementary Fig. 5g, k). These results suggest that ATP7A protects KRAS-mutated cells from Cu toxicity. To verify if oncogenic KRAS augments bioavailable Cu levels in human CRC cells, we evaluated CCS and ATP7A levels in KRAS wild-type (CACO-2) and KRAS-mutated (HCT116, DLD-1, SW480, and SW620) cells. Compared to KRAS wild-type CRC cells, we observed that KRAS-mutated cells had low CCS levels that inversely correlated with high ATP7A levels (Fig. 3f). Consistent with this, we found that KRAS-mutated CRC cells (HCT116 and DLD-1) had a significant increase in fluorescence ratio using crisp-17 compared to wild-type KRAS cells (CACO-2), upon addition of DTDP, suggesting elevated intracellular labile Cu pools in the KRAS-mutant cells (Fig. 3g). Additionally, ICP-MS analysis revealed enhanced total Cu levels in both KRAS-mutated CRC cells compared to CACO-2 (Supplementary Fig. 5l). According to ICP-MS data, HCT116 cells had higher Cu levels than DLD-1 cells, which correlated with the fluorescence intensity ratios in Fig. 3g. To determine if KRAS mutation and ATP7A upregulation correlates with a protective role against Cu toxicity, CRC cells were subjected to increasing concentrations of exogenous Cu. Consistent with data obtained in IEC-6 cells, we found that KRAS-mutated CRC cells had reduced sensitivity to Cu toxicity compared to wild-type counterparts (Fig. 3h, Supplementary Fig. 5m). We evaluated the clinical relevance of these findings by assessing ATP7A and CCS levels in CRC specimens harboring wild-type or mutant KRAS (Supplementary Fig. 6). Interestingly, while total ATP7A levels were similar between the two groups, KRAS-mutated samples were associated with more perinuclear and granular staining of ATP7A suggesting enhanced expression in the Golgi network (Supplementary Fig. 6) and an increase in cell-surface levels of ATP7A (Fig. 3i). We also observed that KRAS-mutated tumors exhibited decreased CCS levels compared to wild-type counterparts (Fig. 3j, Supplementary Fig. 6). Altogether, these data show that oncogenic KRAS enhances ATP7A cell-surface expression, which likely protects KRAS-mutated CRC cells from increased intracellular Cu levels.

**ATP7A influences Cu-dependent tumor growth.** Cu is known to influence tumor growth through the stimulation of several cuproenzymes, including Ceruloplasmin (CP), MEK1/2, and Cytochrome C Oxidase (CCO/COX), which are associated with various biological functions[30–33]. To stimulate these cuproenzymes, Cu is loaded in the secretory pathway via a process that is dependent on ATP7A[34–37]. Strikingly, we found that ATP7A knockdown significantly increased CCS levels in KRAS IEC-6 cells (Fig. 4a), suggesting a reduction in bioavailable Cu levels. Consistent with this, we also observed that ATP7A depletion diminished both CP activity (Supplementary Fig. 7a) and ERK1/2 phosphorylation (Supplementary Fig. 7b). Together, these data suggest that ATP7A modulates Cu bioavailability and contributes to the activity of several cuproenzymes in mutant KRAS-addicted cells.

We then examined whether KRAS-mutated cells may be addicted to enhanced Cu-dependent metabolism using the Cu chelator tetrathiomolybdate (TTM), which is currently being tested clinically against several solid tumors[38,39]. We found that TTM treatment of KRAS cells reversed the increase of ATP7A levels (Fig. 4b, Supplementary Fig. 7c), increased CCS levels (Fig. 4b) and dramatically reduced the mRNA levels of four Cu-dependent genes (Supplementary Fig. 7d), suggesting reduced bioavailable Cu levels. Consistent with this, TTM significantly reduced the activity of Cu-dependent enzymes, such as CP (Fig. 4c) and ERK1/2 phosphorylation in KRAS cells (Fig. 4d, Supplementary Fig. 7e), validating the decrease of bioavailable Cu pools due to Cu chelation. Interestingly, we found that KRAS cells

are 30-times more sensitive to TTM (IC50~1.2 μM) as compared to Control cells (IC50~40 μM) (Fig. 4e). Similar results were obtained comparing mutant KRAS (HCT116, DLD-1) and wild-type KRAS (CACO-2) CRC cells under adherent (Fig. 4f) and non-adherent conditions (Supplementary Fig. 7f), suggesting that oncogenic KRAS increases the requirement for Cu bioavailability[37].

We next examined whether TTM-mediated Cu depletion affected mitochondrial oxidative phosphorylation and concomitant ATP production in KRAS cells. We monitored oxygen consumption rate (OCR) (Fig. 4g), and found that both mitochondrial respiration (Fig. 4h) and ATP production (Fig. 4i) were significantly reduced upon TTM treatment. To determine whether ATP7A contributed to mitochondrial respiration in KRAS cells, we measured ATP levels in KRAS-mutated CRC cells (HCT116 and DLD-1) depleted for ATP7A. Interestingly, ATP7A knockdown significantly decreased ATP levels in these cells (Supplementary Fig. 7g), suggesting that ATP7A participates in Cu-loading of enzymes involved in mitochondrial oxidative phosphorylation. Consistent with these findings, we also found that ATP7A depletion significantly reduced the activity of CCO (Supplementary Fig. 7h) in KRAS-mutated cells, which was reported to correlate with Cu-dependent maximal mitochondrial respiration[33]. Together, our data indicate that KRAS-driven tumor growth is influenced by Cu-dependent mitochondrial respiration, which partly involves the biosynthetic role of ATP7A.

**Macropinocytosis regulates Cu bioavailability in KRAS tumors.** Our data show that oncogenic KRAS increases intracellular Cu levels, but the mechanism by which this occurs remains elusive. While CTR1 is the main Cu-importer in cells[40], our results demonstrate that oncogenic KRAS reduces total and cell-surface CTR1 levels in IEC-6 cells (Supplementary Fig. 8a, b), and in CRC specimens (Supplementary Fig. 8c). In addition, both in vitro (Fig. 2c, Supplementary Fig. 8d) and in vivo (Fig. 2h, Supplementary Fig. 8d) CRISPR/Cas9 screens revealed that CTR1 (*Slc31a1*) is dispensable for KRAS-mutated cell fitness and tumor growth. Recent studies have highlighted the importance of macropinocytosis in nutrient supply for KRAS-mutated cells[41], suggesting a potential alternative route for Cu uptake. Using TMR-Dextran, we confirmed that KRAS cells have enhanced rates of constitutive macropinocytosis compared to Control cells (Fig. 5a), which was inhibited by 5-(N-ethyl-N-isopropyl) amiloride (EIPA), a selective macropinocytosis inhibitor[42]. Interestingly, KRAS cells were found to be more sensitive to EIPA compared to Control cells (Fig. 5b), confirming the acquired dependence of KRAS cells for macropinocytosis. To determine if macropinocytosis regulates bioavailable Cu levels, Control and KRAS cells were treated with increasing concentrations of EIPA, which appeared to preferentially increase CCS levels in KRAS compared to Control cells (Fig. 5c). Furthermore, we found that inhibition of macropinocytosis exhibited a small but statistically significant decrease in the basal fluorescence ratio of crisp-17 in KRAS cells (Fig. 5d), suggesting that macropinocytosis contributes to the supply of bioavailable Cu. Similarly, we found that EIPA treatment augmented CCS levels in KRAS-mutated compared to wild-type CRC cells (Supplementary Fig. 9a). Interestingly, we observed a decrease in ATP7A levels in response to EIPA treatment of KRAS cells (Supplementary Fig. 9b), suggesting that reducing bioavailable Cu levels inhibits ATP7A expression.

To determine if macropinocytosis contributed to Cu uptake and tumor growth in vivo, KRAS cells ($3 \times 10^6$) were xenografted into nude mice subjected to a Cu-deficient diet (Fig. 5e). While group A received regular drinking water, groups B and C were given Cu-supplemented drinking water. Xenografted cells were

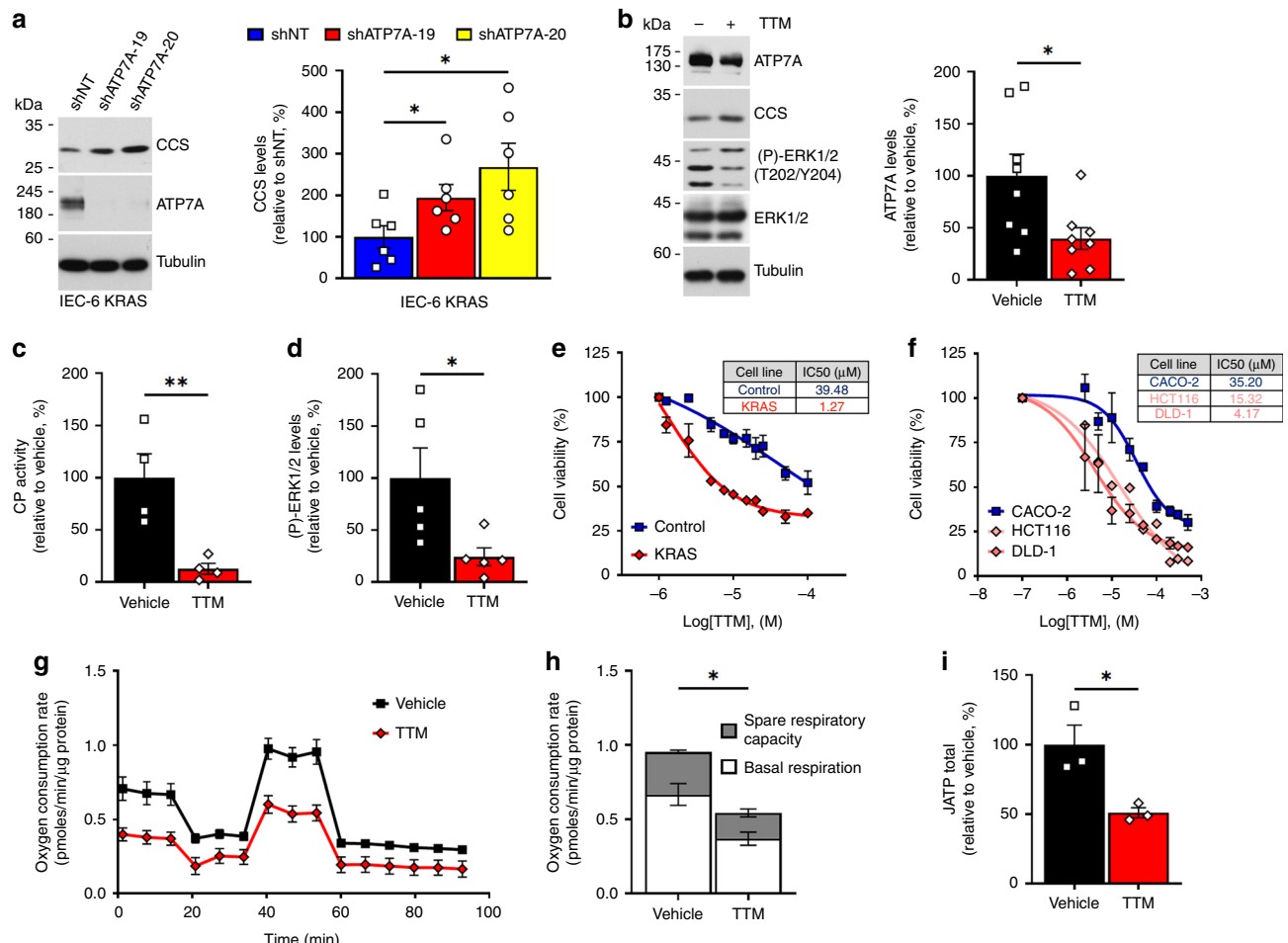

**Fig. 4 Cu chelation reduces Cu-dependent functions associated with tumor growth. a** IB showing CCS levels upon ATP7A knockdown with NT or ATP7A shRNAs (19 and 20) in KRAS cells, with quantifications (right panel). Data represent $n = 6$ independent experiments. *$P = 0.0460$ (shNT versus shATP7A-19); *$P = 0.0223$ (shNT versus shATP7A-20). **b–d** KRAS cells were left untreated or treated with TTM for 24 h and **b** whole-cell lysates were analyzed by IB using antibodies against ATP7A, CCS, total and phosphorylated ERK1/2 and Tubulin. Representative IB images (**b**, left), with quantification of ATP7A levels upon TTM treatment (**b**, right). Data represent $n = 8$ independent experiments. *$P = 0.0230$. **c** KRAS cells were analyzed for Ceruloplasmin (CP) activity. Bar chart indicates relative CP activity and data represent $n = 4$ independent experiments. **$P = 0.0098$. **d** As in **b** but with quantification of phosphorylated ERK1/2 levels upon TTM treatment. Data represent $n = 5$ independent experiments. *$P = 0.0380$. **e** Graph depicting the viability of Control and KRAS cells in the presence or absence of TTM at indicated doses. **f** As in **e** but for CACO-2, HCT116, and DLD-1. Data are presented as relative percentage (±SEM) of living cells and represent $n = 3$ independent experiments with three replicates; IC50 values are indicated in adjacent table (**e**, **f**). **g** Mitochondrial respiration depicted by Oxygen consumption rate (OCR) was measured in real-time in response to mitochondrial inhibitors oligomycin, FCCP, rotenone and monensin. **h** Bar charts representing the mitochondrial basal respiration rates (white), *$P = 0.0249$; the spare respiratory capacity (SRC, gray), *$P = 0.0176$; and **i** cellular ATP production rates, *$P = 0.0278$. **g–i** Data represent $n = 3$ independent experiments with six technical replicates. For panels **a–d**, **g–i**, center values and error bars represent mean ± SEM, and significance was determined using unpaired two-tailed Student's $t$-tests.

allowed to grow for 2 weeks before administering EIPA (group C) or vehicle (groups A and B). As expected, we found that KRAS tumors grew significantly more in Cu-supplemented compared to Cu-deficient conditions (Fig. 5f, g). While Cu deficiency reduced serum CP activity (Supplementary Fig. 9c), mice did not lose weight as a result of the procedure (Supplementary Fig. 9d). Interestingly, we found that KRAS tumors exposed to EIPA grew significantly less compared to vehicle (Fig. 5f, g), suggesting that inhibition of macropinocytosis and Cu uptake negatively affected tumor growth. We did not determine the impact of EIPA treatment in Cu-deficient conditions, as the combination was found to be overly detrimental to the mice. Tumors were analyzed for CCS and ATP7A expression, which revealed that EIPA treatment increased CCS (Fig. 5h) and decreased ATP7A (Fig. 5i) levels. Importantly, decreased Cu levels in EIPA-treated KRAS tumors correlated with a reduction in proliferation, as suggested

by Ki67 staining (Fig. 5j), and increase in necrotic areas (Supplementary Fig. 9f). Finally, we analyzed ERK1/2 and observed an increase of its phosphorylation state in Cu-treated versus Cu-deficient KRAS tumors, which was reduced by EIPA treatment in the presence of exogenous Cu (Supplementary Fig. 9e). Collectively, these data suggest that Cu promotes KRAS tumor growth, and that blocking macropinocytosis reduces Cu uptake, Cu-dependent signaling, as well as tumor growth.

## Discussion
Several studies have demonstrated elevated Cu levels in tumors and sera from cancer patients[43]. Interestingly, Cu levels was shown to correlate with poor prognosis and resistance to chemotherapy[44]. Despite these observations, the mechanisms underlying Cu accumulation and tumor adaptation to excess Cu

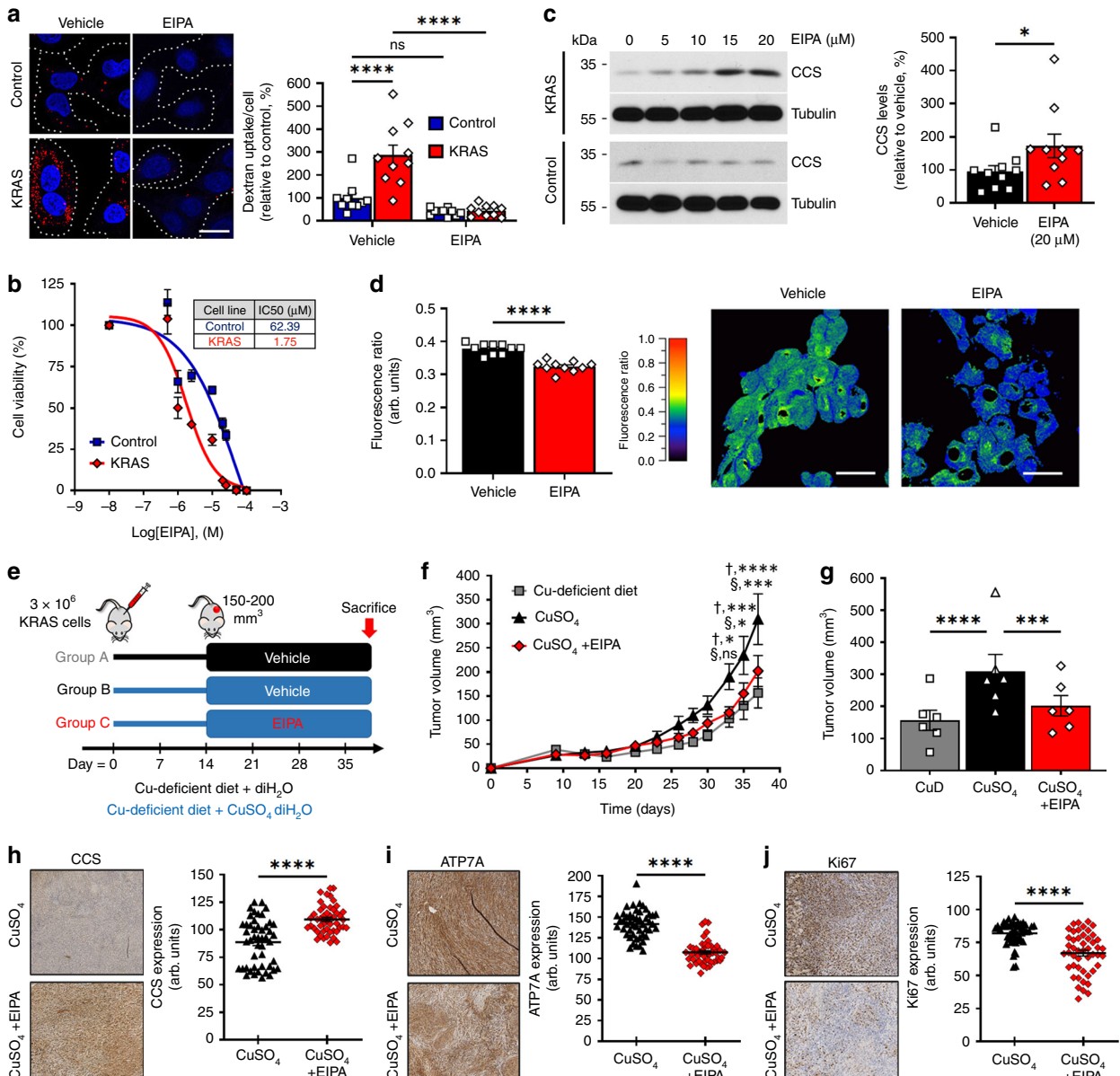

**Fig. 5 Macropinocytosis is a novel Cu supply route in mutant KRAS-driven intestinal cells. a** Macropinocytosis was visualized (left) and quantified (right) with TMR-dextran in Control and KRAS cells. EIPA served as negative control. White dashed lines indicate cell boundaries. Data are averaged for *n* > 167 cells and represent three independent experiments. ****P < 0.0001 (two-way ANOVA with post-hoc Bonferroni's multiple-comparison analysis). scale = 20 μm. **b, c** Control and KRAS cells were treated with EIPA at the indicated doses. **b** Cell viability graph with IC50 values in the adjacent table. Data are presented as relative percentage (±SEM) of living cells and represent *n* = 3 independent replicates. **c** Cell lysates were collected and CCS levels were analyzed by IB and quantified (right). Data represent *n* = 3 (left) and *n* = 10 (right)independent experiments. *P = 0.0336 (unpaired two-tailed Mann–Whitney *t*-test). **d** Graphs depicting average fluorescence ratios of crisp-17 for KRAS cells treated with either DMSO (vehicle) or EIPA (20 μM). Data are averaged for *n* = 10 cells and represent three independent experiments. Representative fluorescence intensity images are presented (right). scale = 30 μm. **e** In vivo protocol used to evaluate the role of macropinocytosis in Cu-dependent KRAS tumor growth. Mice injected with KRAS IEC-6 cells were provided with a Cu-deficient diet and either deionized H2O (diH2O) (Group A) or diH2O with CuSO4 (Groups B and C) throughout the study. After two weeks, groups A and B were treated with vehicle and group C with EIPA for three weeks. Graphs depicting **f**, mean tumor volume over time and **g** mean tumor volume at the end of study (*n* = 6 mice). *P*-values between Group A and B (†, *P < 0.05; ***P < 0.001; ****P < 0.0001) and between Group B and C (§, *P < 0.05; ***P < 0.001) were determined by two-way ANOVA with post-hoc Bonferroni's multiple-comparison analysis. **h–j** Representative images (left) and quantifications (right) from immunohistochemistry profiling of mouse xenograft tissues for *n* = 48 CCS (**h**), ATP7A (**i**), and Ki67 (**j**) random images from six mice per condition are depicted. scale = 250 μm. For panels **a–d**, **f-j** center values and error bars represent mean ± SEM. For panels **c**, **d**, **h–j** *P < 0.05; ****P < 0.0001 (unpaired two-tailed Student's *t*-tests).

remain poorly understood. Previous studies have shown that Cu influx via the Cu-importer CTR1 plays a crucial role in mutant BRAF-driven tumor growth[45]. Our findings indicate that macropinocytosis facilitates Cu entry via a noncanonical route in

KRAS-mutant cells, and that CTR1 plays a negligible role in the growth of KRAS-mutant cells as suggested by the CRISPR screens. Cu likely enters cells in association with its chaperones[46], such as albumin, which was also shown to act as a source of

amino acids for KRAS-mutant cells[47]. While macropinocytosis may play an even larger role in nutrient uptake, our results demonstrate that it provides bioavailable Cu required for the growth of KRAS-mutant tumors. We expect that macropinocytosis may not be the exclusive mode of Cu supply in these cells and other mechanisms such as autophagy or other low-affinity transporters could also play a role[48].

Our results show that oncogenic KRAS increases intracellular Cu levels, which promotes the stability of ATP7A as well as its cell-surface expression. While CTR1 is internalized and degraded in response to high Cu levels, ATP7A translocates to the cell-surface to export excess Cu ions[44]. The latter mechanism is likely responsible for the increased dependency of KRAS-mutant cells towards ATP7A, suggesting that ATP7A inhibitors may specifically target KRAS-addicted cancer cells by exacerbating Cu toxicity. Interestingly, the gastric proton pump inhibitor Omeprazole was shown to inhibit ATP7A surface expression induced by excess Cu and inhibit melanogenesis[49], but drugs that are more specific or function-blocking antibodies against ATP7A may be required to limit treatment-related toxicity.

ATP7A delivers Cu to many cuproenzymes involved in several aspects of tumorigenesis, including cell proliferation, metastasis and angiogenesis[44]. A recent study indicated that ATP7A plays essential roles in loading lysyl oxidase (LOX) and LOX-like (LOXL) proteins, which have well-documented roles in tumor metastasis[37]. Our results suggest that KRAS-mutant cells are addicted to high Cu levels, which is consistent with their increased sensitivity to Cu chelation. While Cu-chelating drugs are being tested against different solid cancers[50–52], our results suggest that KRAS-mutated cancers may be particularly sensitive to such treatments. Cu chelation or ATP7A targeted therapy has not been exploited in CRC, but may be of particularly interest for cases where therapeutic resistance is imparted by oncogenic KRAS.

## Methods

**Cell culture, RNA interference, and viral infections.** IEC-6 and CRC cell lines (CACO-2, HCT116, DLD-1, SW620, and SW480) were from ATCC and maintained at 37 °C in Dulbecco's modified Eagle's medium (DMEM) with 4.5 g/L glucose supplemented with 5% (v/v) fetal bovine serum (FBS), 100 IU/mL penicillin, and 100 μg/mL streptomycin. Cells were regularly tested by PCR to exclude mycoplasma contamination and used within 20 passages. IEC-6 cells were stably transduced with retroviral vectors encoding human KRAS[G12V] or an empty cassette with puromycin, zeocin, or hygromycin selection markers (Addgene #9052, #1764, #1766, #18750). Retroviral particles were produced using the Phoenix cell line and antibiotic selections were performed in 5 μg/mL puromycin, 150 μg/mL hygromycin, or 100 μg/mL zeocin. Small interference RNA (siRNA)-mediated knockdown of ATP7A in IEC-6 cells was achieved by Flexitube siRNA (Qiagen) using Lipofectamine transfection reagent (ThermoFisher Scientific). Briefly, cells were transfected with 7.5 μL Lipofectamine and 75 nM siRNA in Reduced Serum OptiMEM solution, which is replaced 6 h later with regular media and let to grow for 48 h before harvesting for experiments. Short hairpin RNA (shRNA)-mediated knockdown of ATP7A was achieved using lentiviruses produced with vectors from the Mission TRC shRNA library (rat TRCN0000101812, TRCN0000101813, and human TRCN0000043173, TRCN0000043177). Cells were infected in the presence of 4 μg/mL polybrene and were selected 3 days after viral infection with 5 μg/mL (IEC-6), 3 μg/mL (CACO-2), 1.5 μg/mL (HCT116), or 2.5 μg/mL (DLD-1) puromycin.

**Purification and enrichment of cell-surface proteins.** Two proteomic procedures for the enrichment of cell-surface proteins were adapted from previously published methods[53]. For Cell Surface Sapture (CSC), cells were chilled on ice, rinsed twice with ice-cold biotinylation buffer [phosphate buffered saline (PBS), pH 7.4, supplemented with 1 mM CaCl₂ and 0.5 mM MgCl₂], and then incubated with 1 mM sodium (meta)periodate (Sigma-Aldrich) at 4 °C for 30 min in the dark. The mild-oxidation reaction was quenched by addition of glycerol at a final concentration of 1 mM. Cells were then washed twice with ice-cold PBS (pH 7.4) supplemented with 5% (v/v) FBS, and biotinylated for 1 h at 4 °C with a mix of 100 μM aminooxy-biotin (Biotium Inc.) and 10 mM aniline (Sigma-Aldrich) in ice-cold PBS (pH 6.7) supplemented with 5% FBS. Cells were then washed once with ice-cold PBS pH 7.4/5% FBS, and then once with ice-cold biotinylation buffer. Biotinylated cells were incubated in Surfaceome Lysis Buffer [noted as SLB; 1% Triton X-100, 150 mM

NaCl, 10 mM Tris–HCl, pH 7.6, 5 mM iodoacetamide (Sigma-Aldrich), 1× protease inhibitor (cOmplete, without EDTA, Roche), 1 mM sodium orthovanadate (Na₃VO₄), and 1 mM phenylmethylsulfonyl fluoride (PMSF)] for 30 min at 4 °C. Cell debris and nuclei were removed by successive centrifugation for 10 min at 4 °C, initially at 2800 × g and then 16,000 × g. Next, biotinylated proteins were isolated from 10 mg total protein by incubating cell lysates with high-capacity streptavidin agarose resin (Thermo Fisher Scientific) for 2 h at 4 °C. Beads were washed extensively with intermittent centrifugation at 1000 × g for 5 min to eliminate all potential contaminants bound to biotinylated proteins. Three washes were performed with SLB, once with PBS pH 7.4/0.5% (w/v) sodium dodecyl sulfate (SDS), and then beads were incubated with PBS/0.5% SDS/100 mM dithiothreitol (DTT), for 20 min at RT. Further washes were performed with 6 M urea in 100 mM Tris–HCl pH 8.5, followed by incubation with 6 M urea/100 mM Tris–HCl pH 8.5/50 mM iodoacetamide, for 20 min at RT. Additional washes were performed with 6 M urea/100 mM Tris–HCl pH 8.5, PBS pH 7.4 and then water. Biotinylation efficiency was confirmed by blotting aliquots of cell lysates with streptavidin-horseradish peroxidase (HRP) (Dilution 1:50,000). For proteomic analysis, beads were rinsed thrice with 50 mM ammonium bicarbonate (NH₄HCO₃) pH 8.5, and re-suspended in 400 μL of 50 mM NH₄HCO₃ pH 8.5 containing 4 μg of proteomics grade trypsin (Sigma-Aldrich), overnight at 37 °C. The proteins were further digested with an additional 4 μg trypsin for 4 h at 37 °C. The resulting tryptic peptides were then collected by centrifugation at 10,000 × g, for 10 min at RT. The beads were washed twice with MS grade water and the tryptic fractions pooled. The tryptic fractions were dried to completion in a SpeedVac and re-suspended in MS solvent (5% aqueous Acetonitrile (ACN), 0.2% FA). For Cell Surface Biotinylation (CSB), cells were chilled on ice, washed twice with ice-cold biotinylation buffer, and incubated with 1 mg/mL Sulfo-NHS-LC-biotin (resuspended in biotinylation buffer) for 1 h at 4 °C. The biotinylation reaction was quenched by addition of 100 mM glycine for 10 min at 4 °C, followed by two washes with ice-cold biotinylation buffer. Biotinylated cells were lysed in SLB, proteins isolated and digested for MS analysis as described above.

**Mass spectrometry and database searches.** Samples were loaded on a 1.5 μL C18 pre-column (Optimize Technologies) connected directly to the switching valve. They were separated on a homemade reversed-phase column (150 μm i.d. by 150 mm) with a 56-min gradient from 10 to 30% ACN/0.2% FA and a 600-nl/min flow rate on a Ultimate 3000 LC system (Eksigent, Dublin, CA) connected to an Q-Exactive Plus (Thermo Fisher Scientific, San Jose, CA). Each full MS spectrum acquired at a resolution of 70,000 was followed by 12 tandem-MS (MS–MS) spectra on the most abundant multiply charged precursor ions. Tandem-MS experiments were performed using collision-induced dissociation (CID) at a collision energy of 27%. Proteomic samples were analyzed as biological, back-to-back triplicates per condition. Peptides were identified using PEAKS 7.0 (Bioinformatics Solutions, Waterloo, ON) and peptide sequences were blasted against the Rat Uniprot database. Mass tolerances on precursor and fragment ions were 10 ppm and 0.01 Da, respectively. The false discovery rate (FDR) for peptide and protein was set to 0.5%. The minimum number of peptides per protein was set to 2, and minimum peptide length was set to ~6 amino acids. Search criteria included a static modification of cysteine residues of +57.0214 Da; a variable modification of +15.9949 Da to include potential oxidation of methionines; and a modification of +79.966 on serine, threonine, or tyrosine for the identification of phosphorylation. The data were visualized with Scaffold 4.4.6. Normalized Spectral abundance factors (NSAF) for the significantly upregulated or downregulated cell-surface proteins by KRAS (Log2 FC values), for each of the replicates of CSC or CSB technique, were extracted from Scaffold 4.4.6.

**Label-free quantification and data processing.** Peptide precursor intensities were extracted using an integral algorithm of PEAKS® software 8.05. Proteins were then selected based on their detection in at least two replicates of the same biological condition (either Control or KRAS), with a minimum of two unique peptides, which were verified for uniqueness and leucine/isoleucine switch using the neXt-Prot checker[54]. These proteins were considered as identified with high-confidence. Gene Ontology Cellular Component (GO.CC) terms and functional annotations were queried for all identified proteins using g:profiler (http://biit.cs.ut.ee/gprofiler/index.cgi) to get an objective estimation about the number of proteins specific to the cell surface. We then manually sorted all identified proteins such that we kept those that contain at least one cell surface-exposed domain, which could be potentially biotinylated. To do so, we analyzed proteins identified with high-confidence by querying them against the UniProt online database (http://www.uniprot.org/). We then defined proteins as integral to plasma membrane proteins when they contained at least one transmembrane domain and an extracellular region (e.g., CAMs, RTKs), as belonging to secretory or extracellular components when they were known to be secreted and potentially interact with the cell surface (e.g., growth factors, cytokines), and as potential contaminants when they corresponded to proteins that do not possess an extracellular domain (e.g., abundant intracellular proteins that may interact with streptavidin beads, plasma membrane-tethered proteins that interact intracellularly with a biotinylated protein). Proteins were considered differentially upregulated by KRAS if log2 FC (KRAS/Control) values were ≥ 2, and downregulated if ≤ −2. Some proteins were only identified in the KRAS or Control condition, making it impossible to calculate fold-change

values. Therefore, we arbitrarily assigned +15 if proteins were only present in the KRAS condition, and −15 if they were only identified in the Control condition. The value "±15" was chosen as it was higher than our highest and lowest KRAS/Control fold-change, respectively.

**Inductively coupled plasma-mass spectrometry (ICP-MS).** Total Cu levels were measured by ICP-MS. Briefly, adherent cells were cultured in regular complete medium, washed once with PBS prepared in copper-free ddH$_2$O and harvested using Accutase solution (Sigma–Aldrich). Cells were washed once more with PBS and pelleted cells were weighted. All tubes were washed in 1% ultratrace HNO$_3$ (Sigma-Aldrich) to remove excess ions prior to use. Cu measured in the sample tubes is subtracted by background measured in a blank tube, which was treated similar to the sample tubes. Samples were digested in 0.2 mL of concentrated ultratrace HNO$_3$ and 50 μL of 30% H$_2$O$_2$ was added to the weighted mass of solid product. The mixture was then heated at 85 °C for 2 h. After cooling, the volume was adjusted to 10 mL with Milli-Q water prior to analysis in Perkin Elmer Nex-ION 300× ICP-MS instrument. Data are representative of three independent biological experiments.

**SDS-PAGE and immunoblotting (IB).** Cells were washed twice with cold PBS (pH 7.4) and lysed in BLB lysis buffer [10 mM K$_3$PO$_4$, 1 mM EDTA, 5 mM EGTA, 10 mM MgCl$_2$, 50 mM β-glycerophosphate, 0.5% Nonidet P-40, 0.1% Brij 35, 0.1% deoxycholic acid, 1 mM/L Na$_3$VO$_4$, 1 mM PMSF, and complete protease inhibitor cocktail] for 15 min at 4 °C. Lysates were centrifuged at 16,000 × g for 10 min at 4 °C, supernatants were collected and heated for 10 min at 95 °C in Laemmli buffer [50% (v/v) 4× Tris/SDS pH 6.8, 40% (v/v) glycerol, 8% (w/v) SDS, 6.2% (w/v) DTT, 2 mg Bromophenol Blue]. Alternatively, for streptavidin pull-down assays, proteins were biotinylated as described previously and eluted by heating (30 min, 95 °C) in a Laemmli buffer [50 mM Tris-HCl pH 6.8, 5% (w/v) SDS, 5% (v/v) β-mercaptoethanol, 50% (v/v) glycerol, 0.025% (w/v) Bromophenol Blue]. For IB analysis, eluates and total cell lysates were subjected to 8–12% SDS-PAGE, and resolved proteins were transferred onto polyvinylidene fluoride (PVDF) membranes. Membranes were blocked with 10 mM Tris pH 7.4, 150 mM NaCl, and 0.1% Tween 20, supplemented with 5% (w/v) dry skim milk powder, and subsequently immunoblotted with primary antibodies, diluted in 5% (w/v) dry skim milk or 5% (w/v) bovine serum albumin (BSA), overnight at 4 °C. Antibodies targeted against phosphorylated (P)-ERK1/2 (E10) (T202/Y204) (1:1000) is from Cell Signaling Technologies; Tubulin (T5618) (1:2000) from Sigma-Aldrich; Axl (C-20) (1:1000), CCS (H-7) (1:500) from Santa Cruz Biotechnology; Pan-Ras (C-4) from Abcam (1:500); ATP7A (CT77) from Betty Eipper lab (1:1000) and ATP7A (5E-10) from James Collins lab (1:2000). All secondary horseradish peroxidase (HRP)-conjugated antibodies used for immunoblotting were purchased from Chemicon. All IB data are representative of at least three independent biological experiments and were quantified using Image J. Uncropped blots corresponding to the panels presented in the Main or Supplementary Figures are included in the Source Data File, indicating the corresponding figure numbers, molecular weight markers and regions of interest for each protein.

**Cell proliferation and cell counting assays.** Cell proliferation in vitro was assessed colorimetrically with WST-1 reagent. Exponentially growing Control and KRAS IEC-6 cells were grown in 96-well plates in complete medium supplemented with 5% or 0.1% FBS, and the relative number of viable cells was measured at 24, 48, and 72 h. WST-1 was added 2 h prior to absorbance measurement at 450 nm using a Tecan GENios Plus microplate reader. For determination of cell viability, cells were plated in 6-well plates either in complete medium or serum-starved medium, and treated with vehicle [deionized H$_2$O (diH$_2$O) or DMSO], CuCl$_2$ (diluted in diH$_2$O), tetrathiomolybdate [TTM (Sigma-Aldrich), in 5% (v/v) DMSO/diH$_2$O] or 5-(N-ethyl-N-isopropyl) amiloride [EIPA (Sigma-Aldrich), in DMSO] for 24 or 48 h at the indicated doses. Alternatively, cells were left to grow for an additional day in complete medium after ATP7A shRNA selection or siRNA transfection. Cells were harvested using 0.25% Trypsin-EDTA solution (Gibco) and counted with trypan blue (Gibco) to exclude dead cells.

**Anchorage-independent growth assays.** For anchorage-independent growth assays, equal cell numbers were resuspended in 1 mL of top agar solution [final concentration of 0.3% Noble agar (Difco) in DMEM (1×)/10% FBS and antibiotics], with or without TTM at indicated doses. This was overlaid over 1.5 mL of bottom agar solution (final concentration of 0.5% Noble agar in same medium) in a 35 mm non-TC treated culture dish. Cells were fed weekly with 1 mL of complete media, supplemented with indicated concentrations of TTM or vehicle. The number of colonies were visualized after 2 or 3 weeks by addition of 0.5 mg/mL solution of Thiazolyl Blue Tetrazolium Bromide (MTT) reagent (Sigma-Aldrich). Colonies were counted using Image J software.

**Immunofluorescence microscopy.** Cells were seeded in 24-well plates containing coverslips in complete medium for 24 h, then serum-starved for macropinocytosis assay. Lysine-fixable TMR-Dextran 10 kDa (Sigma-Aldrich) was added at 0.5 mg/mL to serum-free media for 30 min at 37 °C, pre-treated with DMSO or EIPA (20 μM). Cells were washed twice in ice-cold PBS pH 7.4 and fixed in 3.7%

formaldehyde/ice-cold PBS for 10 min. Cells were washed thrice in ice-cold PBS, dried and mounted on microscopy slides with DAPI Vectashield (Vector laboratories). For other staining, cells were permeabilized after fixation for 10 min in PBS containing 0.3% Triton X-100 and blocked with PBS/2% FBS or BSA for 1 h. Cells were incubated for 30 min with Texas Red-Phalloidin (Invitrogen) or 1 h with AlexaFluor 488-conjugated Streptavidin (1:500) (Invitrogen) or corresponding primary antibodies. Antibodies targeted against EGFR (C74B9) (1:200), CD44 (8E2) are from Cell Signaling Technologies (1:2000); EphA2 (C-20) (1:25), TLR4 (25) (1:50) from Santa Cruz Biotechnology; E-cadherin (36/E) (1:250), N-cadherin (32/N) (1:400) from BD Biosciences; and SLC31A1/CTR1 (NBP100-402) (1:50) from Novus Biologicals. Following PBS washes, cells were incubated for 30 min with AlexaFluor 488-conjugated secondary antibodies (Invitrogen). Images were acquired on Zeiss LSM-700 confocal laser or GE Healthcare DeltaVision imaging systems with 20× or 40× objectives, and analyzed with Zen software v2. Macropinosome quantifications and cell-areas from differential interference contrast (DIC) images were calculated using Image J software as described elsewhere[47].

**Emission-ratiometric two-photon excitation microscopy.** For the measurement of labile Cu levels, we used the crisp-17 fluorescence sensor in accordance with the published protocol[24]. Briefly, cells were grown on glass bottom culture dishes (MatTek) in complete medium and treated with or without EIPA (20 μM) for 24 h (Fig. 5 experiments). Cells were then incubated with 1 μM crisp-17 in DMEM (without phenol red indicator) for 20 min and imaged at 37 °C under a humidified 5% CO$_2$ atmosphere using a Zeiss LSM 880 two photon laser (excitation at 880 nm and emission simultaneously at 479–536 nm (ch.1) and 611–750 nm (ch.2) band-pass ranges). Ratios were calculated as ch.2/ch.1. After imaging under basal conditions for 5–10 min, dynamic changes in intracellular Cu levels were induced by treating cells with 500 μM 2,2′-dithiodipyridine (DTDP) or 10 μM CuGTSM ionophoric complex. Copper chelation was performed with 50 μM of the high-affinity Cu chelator PSP-2 directly in imaging medium, 20 min after acquisition with CuGTSM, in accordance with the timeframes specified in the original protocol. Changes in fluorescence emission ratios for the region of interest (ROI) corresponding to the cells (cytoplasmic regions) were quantified and analyzed using Image J software[55].

**Flow cytometry and Annexin V staining.** Cells grown on tissue culture dishes were washed with PBS and harvested with Accutase cell dissociation solution (Sigma-Aldrich) for 5 min at 37 °C. Cells (1 × 10$^6$ cells/mL) were then re-suspended in ice-cold PBS pH 7.4/1% FBS (staining buffer) and blocked in ice-cold PBS pH 7.4/5% FBS for 10 min at 4 °C, followed by two washes in staining buffer. Cells were stained with AlexaFluor 488-conjugated Streptavidin (Invitrogen) or primary antibody for 1 h at 4 °C and washed twice with staining buffer. Primary antibodies targeted against TGFβR1/ALK-5 (1:500) is from Novus Biologicals; Integrin β1-Biotin conjugated (Ha2/5) (1:250) from BD Biosciences. Additional staining was done using AlexaFluor 488-conjugated secondary antibody for 30 min at 4 °C. For the quantification of apoptosis, adherent cells along with those in culture media were collected and washed twice with cold PBS. Cells were stained with PE-Annexin V (BD Biosciences), according to the manufacturers' instructions, and analyzed immediately by flow cytometry. Samples were acquired using BD FACS Canto II instrument and BD FACS Diva v8.0.2 software. Data were analyzed using FlowJo v10 software, and represented as geometric mean of fluorescence is represented as Mean Fluorescence Intensity (MFI) or percentage (%) of Annexin V positive cells.

**Real-time quantitative-PCR (qPCR) analysis and RNA-sequencing.** Total RNA was extracted using RNeasy mini Kit (Qiagen) and reverse-transcribed using the cDNA Reverse Transcription Kit (Applied Biosystems), as described by the manufacturer. Gene expression was determined using assays designed with the Universal Probe Library from Roche (www.universalprobelibrary.com). For each qPCR assay, a standard curve was performed to ensure that the efficacy of the assay is between 90 and 110%. The Viia7 qPCR instrument (Life Technologies) was used to detect amplification level and was programmed with an initial step of 20 s at 95 °C, followed by 40 cycles of: 1 s at 95 °C and 20 s at 60 °C. Relative expression (RQ = 2$^{-\Delta\Delta CT}$) was calculated using the Expression Suite software (Life Technologies), and normalization was done using both *GAPDH* and *ACTB*. Data represent average of two technical replicates from three independent biological experiments. For transcriptome analysis, total RNA was extracted using RNeasy mini Kit (Qiagen). Presence of contamination was assessed by NanoDrop (ThermoFisher Scientific) using 260/280 and 260/230 ratios. Quantification of total RNA was made by QuBit (ABI) and 500 ng of total RNA was used for library preparation. Quality of total RNA was assessed with the BioAnalyzer Nano (Agilent) and all samples had a RIN above 9.8. Library preparation was done with the KAPA mRNA-seq stranded kit (KAPA, Cat no. KK8420). Ligation was made with 9 nM final concentration of Illumina index and 10 PCR cycles was required to amplify cDNA libraries. Libraries were quantified by QuBit and BioAnalyzer. All libraries were diluted to 10 nM and normalized by qPCR using the KAPA library quantification kit (KAPA; Cat no. KK4973). Libraries were pooled to equimolar concentration. Sequencing was performed with the Illumina Hiseq2000 using the Hiseq Reagent Kit v3 (200 cycles, paired-end) using 1.8 nM of the pooled library. Around 120 M

paired-end PF reads were generated per sample. Library preparation and sequencing was made at the Institute for Research in Immunology and Cancer's (IRIC) Genomics Platform. Sequence data were mapped to the reference genome using the Illumina Casava 1.8.1 package and Refseq release 63. Expression levels of mRNA were displayed as reads per kilobase per million (RPKM), these RPKM values were used to analyze the functional enrichment of genes associated with the sequencing data by gene set enrichment analysis (GSEA). Data are representative of three independent biological experiments. RPKM values of each biological replicate were averaged for Control and KRAS conditions, and Log2 FC (KRAS/Control) values were calculated. Transcripts were considered differentially upregulated by KRAS if log2 FC (KRAS/Control) values were ≥2, and downregulated if ≤ −2 ($P < 0.05$). The heat map in Fig. 1c was generated by comparing the list of genes significantly and differentially expressed in KRAS-altered compared to KRAS-unaltered CRC specimens isolated from TCGA (594 samples) using cBioPortal platform (https://www.cbioportal.org) to the list of genes differentially expressed by KRAS in IEC-6 cells, as described above. The resulting list of common genes was further shortlisted on the genes whose expression was previously published in the literature as KRAS dependent. Primer sequences for qPCR analysis are provided in the Supplementary Table 1.

**Immunohistochemistry (IHC).** IHC staining was carried out on paraffin-embedded formalin-fixed samples using the automated Bond RX staining platform from Leica Biosystems. Sections were deparaffinised inside the immunostainer and antigen recovery was conducted using Leica Biosystems proprietary heat-induced Epitope Retrieval using low pH buffer (ER1) for 20 min. Sections were then incubated with 150 μL of each antibody at RT for 30/15 min (primary/secondary antibody, respectively). Primary antibodies targeted against Caspase-3 (1:100) and Ki-67 (1:100) are from Biocare Medical; CCS (H-7) (1:50) from Santa Cruz Biotechnology; SLC31A1/CTR1 (NBP100-402) (1:50) from Novus Biologicals; and (P)-ERK1/2 (E10) (1:400) (T202/Y204) from Cell Signaling Technologies. Detection of specific signal was acquired by using Bond Polymer DAB Refine kit (#DS9800, Leica Biosystems) or Bond Intense R Detection System (DS9263, Leica Biosystems) with a secondary Biotin-conjuguated antibody. Slides were counter-stained automatically with Hematoxylin included in the detection system. Stained slides were scanned using the Hamamatsu's NanoZoomer Digital Pathology system 2HT. Virtual slides were then imported in Visiopharm Integrator System (2019.02.1.6005). Quantification was done by defining random zones of region of interest (ROI) (≥50 per condition), and determining the global intensity of DAB staining or necrotic tissue area by a custom scoring protocol by Visiopharm algorithm using HDAB-DAB color deconvolution and intensity parameters. Within the ROI, an unsupervised k-means cluster analysis was performed to separate pixels into four classes corresponding to negative, low, moderate and strong IHC staining. Mean Intensity (MI) and Area were defined for each class and a global VIS score (VS) was calculated as: VS = (MI × area) LOW + (MI × area) MOD + (MI × area) STRONG / total area of ROI. Other custom applications (APPs) were used from Visiopharm APP center: Ki-67 APP (90004) to determine Ki-67 positive cells, HER-2 APP (90007) and GLUT-1 APP (10009) for ATP7A cell-membrane localization.

Staining of patient-derived CRC tissues was additionally scored by pathologist Dr. Louis Gaboury (Université de Montréal). Data are representative of six and five independent biological replicates for xenografts and patient tissues, respectively. In the present study, ten tissue blocks were retrieved from the archives of the pathology lab at Center Hospitalier de l'Université de Montréal (CHUM), each containing colorectal cancer that also undergone routine testing for RAS mutation. Tumor-containing tissues with sufficient intact material were randomly selected and anonymized to include five KRAS mutated cases and five KRAS wild-type cases. Five to ten unstained sections (2 μm thick) were prepared and donor blocks returned to the archives. On the one hand, each case was anonymized (or depersonalized) to maintain confidentiality, that neither age, sex, ethnicity nor any other personal information was divulgated. On the other hand, care was taken to ensure that by using these tissues there is no potential risk or benefits to the patient other than the advancement of science and knowledge on the cellular mechanisms leading to colonic cancer. Additionally, all patients signed a general consent form upon admittance authorizing the establishment to either dispose or use for research purposes of the tissues and organs removed, so an Institutional Review Board was not deemed necessary.

**Mitochondrial bioenergetics analysis.** Oxygen consumption rate (OCR) was measured using the XFe96 Extracellular Flux Analyzer (Seahorse, Bioscience) according to the manufacturer's protocol. Briefly, KRAS IEC-6 cells were washed twice and incubated in 200 μL of XF media containing 10 mM glucose, 2 mM glutamine, and 1 mM sodium pyruvate for 1 h at 37 °C in a CO$_2$-free incubator. Following the measurement of basal respiration, injections included oligomycin (1 μM), FCCP [carbonyl cyanide 4-(trifluoromethoxy)phenylhydrazone] (0.75 μM), rotenone/antimycin A (0.5 μM), and monensin (20 μM). Oligomycin inhibits ATP synthase and blocks respiration coupled to ATP production (coupled respiration). Uncoupled respiration is calculated as the difference between basal respiration and coupled respiration. FCCP uncouples the inner mitochondrial membrane, thereby allowing for the measurement of maximal oxygen consumption and maximal oxidative capacity. The addition of rotenone (complex I inhibitor) and antimycin A

(complex III inhibitor) are used to maximally perturb mitochondrial respiration. Monensin stimulates Na$^+$ and H$^+$ cycling across the plasma membrane, which activates glycolysis and is used to calculate maximum glycolytic capacity. OCR, extracellular acidification rate (ECAR), and proton production rate (PPR) measurements were taken before and after each injection and were used to calculate J ATP production and bioenergetic capacity as previously described[56]. All values were normalized to protein content.

Alternatively, total ATP was measured using the ATP luminescence assay kit (Abcam). Briefly, cells were grown in complete medium supplemented with 5% FBS for 24 h, lysed and prepared according to the manufacturers' instructions prior to luminescence measurement using a Tecan GENios Plus microplate reader. All values were normalized to cell numbers. For the determination of Cytochrome C Oxidase (CCO) activity, KRAS cells were grown in complete medium supplemented with 5% FBS prior to incubation with 100 μM Cu for 24 h and mitochondrial extracts were prepared with the MITOISO1 mitochondria isolation kit (Sigma) according to the manufacturers' instructions. Cytochrome c oxidase activity was then measured using the CYTOCOX1 kit (Sigma) according to the manufacturers' protocol. All values were normalized to cell number.

**In vivo xenotransplantation into nude mice.** Throughout the study, 18 female athymic nude mice ($n = 6$ per condition) (Charles River stock #0490) around 8 weeks of age were subjected to a copper-deficient chow (CuD diet, TD.80388, Harlan Teklad), with either regular drinking diH$_2$O (Group A) or supplemented with 20 mg/L CuSO$_4$ diH$_2$O (Sigma-Aldrich) (Group B and C). KRAS IEC-6 cells underwent screening to confirm that the cell line was free of rodent infectious agents (Mouse essential clear panel from Charles River), and 3 million cells were resuspended in PBS and injected subcutaneously in the right flank of mice at the beginning of the study. Tumors were allowed to grow until the volume reached ~150–200 mm$^3$ after 2 weeks into the study. Mice were randomized and administered the following treatments every 2 days: Group B vehicle (5% DMSO in diH$_2$O) or Group C EIPA (20 mg/mL in 100 mL of vehicle) by intraperitoneal injections (i.p.) for 3 weeks. All animal procedures were performed in accordance to local animal welfare committee of the Université de Montréal (Comité de Déontologie en Expérimentation Animale, CDEA) in agreement with regulations of the Canadian Council on Animal Care (CCAC). Mice were monitored periodically; tumor volume thrice per week and bodyweight once a week were recorded. Blood was collected from the saphene at the start and end of the study for Ceruloplasmin activity measurements. Mice were sacrificed at the end of the study and observed for macroscopic abnormalities, if any, and tumor tissues were excised and fixed immediately for IHC analysis. Mice were maintained under the following housing conditions: ambient temperature 22 °C, humidity control 50%, 12 h light/ 12 h dark cycle.

**Ceruloplasmin activity assays.** Blood was collected from mice ($n = 6$ per condition), allowed to clot and serum was used for assays on D35. Pre-treatment (D0) sera and TTM-treated sera were used as positive and negative controls respectively. Ceruloplasmin (CP) activity was assayed and calculated according to ref. [57] and adapted[45] for 96-well plates. Briefly, 3.75 μL serum was added to 37.5 μL 0.1 M sodium acetate pH 6.0 (Sigma-Aldrich) in 96-well plates and incubated at 37 °C. After 10 min, 15 μL of 2.5 mg/mL o-dianisidine dihydrochloride (Sigma-Aldrich) was added and further incubated for another 30 min at 37 °C. The reaction was stopped by adding 150 μL of 9 M sulfuric acid (Sigma-Aldrich). The absorbance was read at 540 nm with appropriate negative controls water (blank) and sample (without sodium acetate and o-dianisidine dihydrochloride). For in vitro assays, cells were treated with complete medium containing 50% FBS with or without TTM (10 μM) for 72 h. Conditioned media was collected and concentrated with Amicon Ultra-4 30MWCO (Sigma-Aldrich). 50–100 μL concentrates were used for CP activity measurements, with the same protocol as above but with 750 μL of sodium acetate, 200 μL of o-dianisidine dihydrochloride for 120 min and reaction quenched by adding 2 mL of sulfuric acid. Activity was normalized to total protein content.

**CRISPR/Cas9-based screens.** Lentiviral gRNA library construction: genes coding for cell-surface proteins were selected based on their significant upregulation or downregulation (Log2 FC ≥ +2 or −2) at the surface of KRAS-mutated IEC-6 cells compared to wild-type counterparts. Rat gRNAs against selected cell-surface protein-coding genes were pooled with gRNAs against previously described reference genes that are known to be essential or nonessential to cell survival. This library, collectively referred to as KRAS-library (3732 gRNAs in total, Supplementary Data 4), was synthesized as 58-mer oligonucleotides and amplified by PCR in a pooled format. PCR products were then purified using the QIAquick nucleotide removal kit (Qiagen) and cloned into a modified version of the all-in-one lentiviral vector lentiCRISPRv2 using a one-step digestion and ligation reaction as previously described[58]. The pooled ligation reaction was purified using the QIAquick nucleotide removal kit, eluted in 23 μL ultrapure diH$_2$O, and 2 μL of the purified ligation was transformed into 25 μL EndureTM competent cells (Lucigen) and selected using ampicillin (100 μg/mL). A total of four identical transformations were performed to yield 5000-fold representation of the library. KRAS-library plasmid DNA was harvested using the EndoFree Plasmid Maxi kit (Qiagen).

*Lentivirus production and MOI testing*: KRAS-library lentiviruses were produced by co-transfection of lentiviral vectors psPAX2 (4.8 µg) and pMDG.2 (3.2 µg) with KRAS-library plasmid DNA (8 µg) using the X-tremeGene TM 9 transfection reagent (Roche), as previously described[58]. Briefly, $9 \times 10^6$ HEK293T cells were seeded per 15 cm plate 24 h prior to transfection in 10% FBS in DMEM medium, then transfected with the above-mentioned transfection mixture according to the manufacturer's protocol. Twenty-four hours after transfection, the medium was removed and replaced with fresh medium (DMEM, 1% BSA, 1% Penicillin/Streptomycin). Forty-eight hours of post-transfection, virus containing medium was harvested, filtered (0.45 µm), and stored at −80 °C. KRAS-library viral titers in pWZL-hygro IEC-6 or pWZL-hygro-KRAS[G12V] cells was determined by infecting the cells with a titration of KRAS-library lentivirus in the presence of polybrene (8 µg/mL). Twenty-four hours after infection, medium was replaced with fresh media containing 4 µg/mL puromycin and selected for 48 h. The multiplicity of infection (MOI) was determined 72 h post-infection by comparing the percent of survival of infected cells to non-infected control cells[58].

*KRAS-library CRISPR/Cas9 screening*: 21 million pWZL-hygro IEC-6 or pWZL-hygro-KRAS[G12V] cells were infected with KRAS-library lentivirus at a MOI ~0.3 and selected with puromycin (1500-fold coverage after selection). Seventy-two hours of post-infection, selected cells were harvested, mixed and split into three replicates of 2 million cells to maintain 500-fold library coverage. Cells were passaged every 3–4 days as needed, and 4–5 million cells were collected for genomic DNA extraction at day 0 (D0), day 7 (D7), and day 14 (D14). Genomic DNA was extracted from cell pellets using the Purelink genomic DNA mini kit (Invitrogen). Integrated library gRNAs were amplified via two stage PCR: PCR-1 to enrich gRNA regions in the genome and PCR-2 to amplify gRNA with Illumina TruSeq adapters with i5 and i7 indices[59]. Three 50 µL PCR-1 reactions (1 µg gDNA each) were performed for each genomic DNA sample and then pooled. For PCR-2, 5 µL of pooled PCR-1 were amplified, using unique i5 and i7 index primer combinations for each individual sample in 50 µL reactions. PCR product was separated on 2% agarose gel, and 200 bp bands and excised and purified using the PureLink™ Quick Gel Extraction and PCR Purification Combo Kit (Invitrogen) eluting in 30 µL volume. Recovered DNA was quantified using a Nanodrop and 2 µL was electrophoresed on a 2% agarose gel to ensure a clean 200 bp band was recovered. Next-generation sequencing was performed at the Lunenfeld–Tanembaum Research Institute Sequencing facility as previously described[59]. Libraries were sequenced to a minimum depth of $1 \times 10^6$ reads per sample on a Illumina NextSeq550 and FASTQ files were aligned to the gRNA library using MaGeck (version 0.5.3) using default parameters. For each gene in the screen, experimental condition (Control and KRAS), and timepoint, Bayes factors (BF) were calculated using the BAGEL algorithm including the 100 training non-essential and essential genes present in the library[20]. BF represents a confidence measure and considers the FC of all gRNAs targeting the same gene. A positive BF signifies that the gene affects cell fitness and negative BF signifies that the gene is dispensable to cell fitness. BF were normalized across samples within each screen using the preProcessscore R package before comparison. Differential essentiality scores were calculated after subtracting the BF of KRAS from Control and presented as or Log2 FC. If Log2 FC ≥ 2, genes were considered differentially essential to KRAS cells. Only genes with positive BF were considered for differential essentiality. The results from two in vitro screens using two pairs of isogenic IEC-6 cells with/without mutant-KRAS were combined to represent the mean differential essentiality scores in Fig. 2c and Supplementary Data 5.

*In vivo CRISPR/Cas9 screening*: pWZL-Hygromycin-KRAS[G12V] IEC-6 cells (21 million cells) were transduced at a MOI ~0.3 with the KRAS-library gRNAs, and transduced cells were selected with 4 µg/mL puromycin for 2 days. A D0 sample was collected on the following day. NOD/SCID/gamma (NSG) mice ($n = 9$) were injected with 1 million cells each (250,000 cells × 4 injections), mixed 1:1 with matrigel containing growth factors (Corning #354234). The resulting 36 tumors were harvested after 4 weeks (D30) when they reached ~1 cm³ and assigned randomly to 3 × 12 tumors. DNA from 12 tumors was purified (Qiagen DNEasy Blood and Tissue kit) and pooled to obtain a ~600× coverage/triplicate, which was compared to the 600× D0 sample. Essentiality scores were determined from the BF.

**Statistical analyses**. For surfaceome and transcriptome analysis, an unpaired one-tailed Student's *t*-test was applied to the mean log2 FC value and selected as differentially expressed if $P < 0.05$. For all relevant panels unless specified, center values and error bars represent means ± SEM and P-values were determined by unpaired two-tailed Student's *t*-tests. ns: not significant; $*P < 0.05$, $**P < 0.01$; $***P < 0.001$; $****P < 0.0001$. For Figs. (2e, f, 3e, h, 5a), and Supplementary Figs. (5k, m, 7f), two-way ANOVA with post-hoc Bonferroni's multiple-comparison analysis were used between groups. ns: not significant; $**P < 0.01$; $***P < 0.001$; $****P < 0.0001$. In Fig. 5f, g with mean tumor volume (mm³) ± SEM, P-values between Group A and B (†) F(DFn,DFd) = 30.55(11,126) and between Group B and C (§) F(DFn,DFd) = 37.79(11,127) were determined by two-way ANOVA with post-hoc Bonferroni's multiple-comparison analysis. Coefficient correlation ($R^2$) were calculated in Excel by comparing the means (of three replicates) of log2 (FC values) for CSC versus CSB (Supplementary Fig. 2e), or transcriptome versus surfaceome (Supplementary Fig. 3a). Pearson correlation coefficient ($\rho$) and NSAF values (Supplementary Fig. 2d) were extracted from Scaffold database. Statistical measurements were made on distinct samples using both Graph Pad PRISM v8.4.2 and Microsoft Excel softwares.

**Reporting summary**. Further information on research design is available in the Nature Research Reporting Summary linked to this article.

## Data availability

The raw datasets of surfaceomics (Supplementary Data 2 and 3), transcriptomics (Supplementary Data 1 and 2) and CRISPR analyses (Supplementary Data 4–6) are available as supplementary information in excel sheet formats. Surfaceomics data (corresponding to Supplementary Data 2) has been deposited in ProteomeXchange through partner MassIVE as a complete submission and assigned MSV000085560 and PXD019625. The data can be downloaded from ftp://MSV000085560@massive.ucsd.edu. TCGA data used for generating heat map in Fig. 1c and Kaplan–Meier plot in Supplementary Fig. 1a are available from cBioportal (https://www.cbioportal.org/study/summary?id=coadread_tcga_pan_can_atlas_2018). GSEA graphs presented in Fig. 1b and supplementary data Fig. 1e are available at https://www.gsea-msigdb.org/gsea/msigdb/cards/HALLMARK_KRAS_SIGNALING_UP and https://www.gsea-msigdb.org/gsea/msigdb/cards/HALLMARK_MYC_TARGETS_V1, respectively. All other data are available in the Article or Supplementary Information or available from the authors upon reasonable request. Source data are provided with this paper.

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

## Acknowledgements

We thank all lab members who contributed in discussions and manuscript preparation. Proteomic analyses were performed by the Center for Advanced Proteomics Analyses (CAPCA), a Node of the Canadian Genomic Innovation Network that is supported by the Canadian Government through Genome Canada. IEC-6 cells were provided by Dr. Nathalie Rivard (Université de Sherbrooke), and pWZL-Hygromycin vector by Dr. Gerardo Ferbeyre (Université de Montréal). ATP7A polyclonal antibody was a gift from Dr. Betty Eipper (University of Connecticut) and Dr. James Collins (University of Florida). We acknowledge Julie Hinsinger for the analysis of immunohistochemistry results. We are thankful to Christian Charbonneau and Jennifer Huber for their expertise with bio-imaging techniques and transcriptome analysis, respectively. We are grateful for the time and expertize of Kim Levesque, Myriam Métivier-Bélisle and other personnel at the IRIC animal facility for mouse xenograft experiments. We benefited from the personnel at IRIC's Genomics Core Facility for RNA experiments. This work was funded by grants from the Canadian Institutes for Health Research (MOP-142374 and PJT-152995 to P.P.R), and the National Institutes of Health (R01GM067169 and R35GM136404 to C.J.F). P.P.R. is a Senior Scholar of the Fonds de la recherche du Québec—Santé (FRQS). L.A. received a Postdoctoral Fellowship from the Cole Foundation, and N.N. was supported by Doctoral Scholarships from the FRQS and the Fonds de la recherche du Québec—Nature and Technologies (FRQNT).

## Author contributions

L.A. and N.N. conceived, performed and analyzed most of the experiments, and contributed with P.P.R. to the study design and writing the manuscript. G.L. carried out all retroviral infections, performed WST-1 and Annexin V assays, and helped in confirming qPCR and IB experiments. S.N. performed and analyzed IF experiments, and participated in discussions related to the overall goals of the project. Z.S., J.B., S.L., G.M., and T.H., designed, performed and analyzed CRISPR/Cas9 screens, supervised by S.A. and D.S. Both M.S.E. and S.M. supervised in vivo xenograft experiments, and C.J.F. helped in the design and analysis of experiments using Cu-sensing probes. D.P. performed, designed and analyzed bioenergetics experiments, under I.T. supervision. L.G. supervised IHC experiments and analyzed patient-derived data. S.A., S.M., D.S., and I.T. critically read the manuscript. P.P.R. supervised and conceived the study.

## Competing interests

The authors declare no competing interests.
