## [Peer Review File · Nature Communications]

Reviewers' comments:

Reviewer #1 (Remarks to the Author): Expert in cell surface proteomics

The authors undertake a major proteogenomic study where they demonstrate that oncogenic KRASG12V expression reprograms the intestinal cell transcriptome in a manner that partially recapitulates the surfaceome (cell surface proteome).

They subsequently identify several potential therapeutic targets through cell surface-based (CRISPR/Cas9 screen using a surfaceome-based lentiviral gRNA library) loss-of-function screens. In particular, ATP7A (a copper-export protein) upregulated by mutant oncogenic KRASG12V was also upregulated at the surface of KRAS-mutated colorectal cancer cells and protects them from excess copper-ion toxicity. KRAS-mutated cells required higher Cu levels to support growth, which was found to be provided via a CTR1-independent uptake mechanism involving macropinocytosis.

The authors contend that oncogenic KRAS enhances Cu-dependent ATP7A cell-surface expression, which becomes a vulnerability and potent synthetic lethal target for KRAS-addicted CRC and that KRAS-selective vulnerability may be exploited therapeutically for treatment of KRAS-mutated cancers.

Comments/Improvements:

1. The manuscript is generally well written, logical, easy to follow and describes a well-constructed multi-omics approach.
2. The manuscript would improve considerably if its strength were emphasised. This is a wonderful example of using proteogenomics (i.e., KRASG12V mediated transcriptomics with cell surface proteomics changes) to elucidate novel potential therapeutic targets through cell surface-based loss-of-function screens, in vitro and in vivo studies using Cu supplementation or chelation and comparisons with mutated CRC cell lines and CRC tissues.
3. Technical levels of stringency employed to quantitatively measure specific mRNA and protein level changes in the KRASG12V-mediated transcriptome and cell surface proteome must be explained/enumerated in much greater detail. For example, a study of this importance would be expected to rely on high-stringency proteomics data - whereby the number of peptides used should be 2 or more, the length of non-nested peptides used should be 9 amino acids or more, FDRs should be <1% for both the peptide and protein FDRs, leucine/isoleucine switching should be checked along with use of the neXtProt peptide untypicity checker to ensure peptides used for a protein call are unique-expressed for any given protein.
4. Although their data from CSB and CSC chemoproteomics approaches compliment each other, the reason why both were employed in the is not self-evident - this could be clarified.
5. Again, a rationale for why label-free quantification by liquid chromatography coupled with tandem mass spectrometry (LC-MS/MS) and not TMT-labelling was employed would have been helpful.
6. The involvement of ATP7A in mutant KRAS cell fitness was confirmed by RNAi using IEC-6 cells and the CRC cell lines HCT116 and DLD-1 cells. Whilst the mutational variation between lines HCT116 and DLD-1 cells is well known, the reason for initial choice of the IEC-6 model as the most appropriate could be further explained.
7. Equally, to verify oncogenic KRAS augments Cu levels in human CRC cells, they evaluated CCS and ATP7A levels in KRAS wild-type (CACO-2) and KRAS-mutated (HCT116, DLD-1, SW480, SW620) cells and observed that KRAS-mutated cells had low CCS levels inversely correlated with high ATP7A levels. Explain why they chose to use so many different CRC cell lines here?
8. The effects of using a Cu-supplement diet and chelators are convincing. Did they consider using proteogenomics to determine off-target effects that may have been induced by these treatments?

Reviewer #2 (Remarks to the Author): Expert in macropinocytosis and ras-driven cancer

Review of manuscript "Copper bioavailability is a KRAS-specific vulnerability in colorectal cancers"
In the submitted study, Aubert et al. provide evidence that overexpression of mutant KRAS is sufficient to modulate the expression of many cell-surface proteins, both at an mRNA (the transcriptome) and protein (surfaceome) level. Using a CRISPR/Cas9 loss-of-function screen, the authors identify cell-surface proteins that contribute to the fitness of mutant KRAS cells. From here, the work shifts to focus on one of their most promising hits, ATP7A, a copper exporter. Using protein levels as a biomarker of intracellular copper concentrations, the authors go on to show that mutant-Ras colorectal cell lines are more sensitive to copper chelation, and that indeed, levels of both ATP7A and CCS, copper chaperone for superoxide dismutase, vary with mutant-Ras status in a way that links Ras activity to increased intracellular copper levels. Upon increases in intracellular Cu, ATP7A will be upregulated to ensure sufficient export to prevent Cu toxicity. CCS in turn will be degraded in the presence of high Cu concentrations. Somewhat paradoxically, however, the authors also show that CTR1, the primary copper importer in cells, is actually downregulated, which they state implies that Cu must be entering the cell by other means in the Ras mutant setting. The authors follow up this in vitro and in silico work with an in vivo loss-of-function screen to validate their findings and conclude by showing that macropinocytosis may facilitate copper uptake into Ras-mutant colorectal cancer cells in lieu of CTR1 levels being downregulated, contributing to their overall fitness. Inhibiting macropinocytosis by treating tumors with EIPA can prevent the rescue of tumor growth by CuSO₄ supplementation in the Cu-deficient setting. The paper looks to be the first to undertake a full loss-of-function screen (both in vitro and in vivo) to identify Ras-dependent changes to cell surface proteins, whereas previous attempts have focused on mutational analysis (PMID: 25193853). Taken as a whole, the submitted work offers a promising glimpse into what may be a novel metabolic vulnerability in Ras-mutant colorectal cancer mediated by cell surface ATP7A, which would be of interest to both the Ras biology and oncology fields. While there is some work to indicate that copper is critical in certain cancers subtypes (PMID: 30824611), this is the first to link this vulnerability directly to mutant Ras and such an advance would be a welcome addition to the aforementioned fields. However, multiple of the authors' claims are left under supported, primarily in the second half of the paper in which a dissection of the importance of copper import is argued, but intracellular copper concentrations are never directly measured. Finally, the argument that "macropinocytosis is required for copper entry into KRAS tumors" is left severely underdeveloped and in need of significantly more data and critical control experiments to support this conclusion. Specific comments are now listed below.

Major Comments

1. The authors argue that ATP7A is a potent synthetic lethal target for KRAS mutant CRC cells. To prove this claim, the authors would be required to show that ATP7A knockdown in the context of non-Ras mutant lines has little to no effect, whereas the lethality in Ras mutant cells will be very high. We note that the target itself is derived from a synthetic lethal CRISPR screen, but nonetheless this hit should be validated in a thorough way. While the data in Extended Data Figure 4e and 4f show that siRNA targeting of ATP7A limits cell doubling in Ras-mutant cells, they do not provide any evidence that this effect is not also seen in Ras-WT cells (as they have for copper supplementation in Figure 2g).

2. While the entire paper revolves around the premise that high intracellular Cu levels indicate a novel metabolic vulnerability in Ras-mutant cells, the authors only use indirect measures (albeit well established in the current literature) of Cu levels in the paper. Not one experiment measures the concentration of copper itself. This is especially glaring in the final figure of the main body of the text, in which the authors propose that macropinocytosis is a key import mechanism for Cu in Ras-mutant cells. To argue this claim, data would be required showing definitively that copper levels drop to extremely low levels in the presence of a macropinocytosis inhibitor.

3. The in-vivo macropinocytosis experiment (Figure 4e) excludes a vital control. Since, as has been previously established in published work (PMID: 23665962) and as the authors acknowledge directly in Figure 4b, Ras-mutant cells are selectively sensitive to inhibition of macropinocytosis by EIPA when compared to non-mutant controls. This necessitates that for their in vivo experiment they add at least one more arm containing EIPA treatment alone. The effects seen from the CuSO₄ to the CuSO₄ + EIPA arm may have little to do with preventing copper uptake and may actually

occur due to the deleterious effects of EIPA independent of this (likely amino acid deprivation). Also, this figure offers little insight into the selectivity of these effects for Ras-mutant cells *in vivo*, as only a mutant-Ras line is tested.

Minor comments

1. In Figure 1c, it is unclear how the gene set was generated. In the text, they write "[...]we found a very good correlation between our dataset and oncogenic KRAS-specific changes occurring in CRC specimens from The Cancer Genome Atlas[...]," however it is not clear if these are simply all the genes differentially regulated by KRAS that are common between the data set or the most statistically significant hit or if this data set was created by some other criteria. Some clarity here would be helpful for the reader.
2. Figure 2c, and 2e; extended data figure 3c (TLR4 graph); extended data 4a, 4b (surfaceome bar), 4e, and 4f; extended data figure 6a, and 6i; and extended data figure 7d are missing error bars..
3. For consistency, all single color images should be presented in black and white, as the authors have done in extended data figure 3b.
4. In extended data Figure 4i, j, the authors present cell death fold change, whereas in Figure 2g, i they present % cell number, and in extended data figure 6b, c, and e, they present transformed growth %. Readouts for cell fitness should be consistent throughout.
5. In Figure 3e bar graphs, what precise treatment group is being quantified?
6. In extended data figure 4h, authors should include the control cell images and specify which images are control and which are ras-mutant cells.
7. Figure 4c bar graph should be consistent with color scheme throughout the whole paper. Vehicle bar should be colored blue or black, as authors have done throughout.
8. Quantification of IHC data is artificially overpowered by including each counted field as an individual data point. The statistical power from such a study is derived from the number of mice used, not the number of images analyzed. Here, n values should be the number of mice on each treatment arm. We recommend that the authors take the average quantification of the images from each mouse and provide that as one value, deriving the error instead from the values of each mouse, leaving n = 6. While this will make achieving statistical significance more difficult, it is a more accurate way to present the data since it is unclear if the values driving results are all derived from outlier mice.
9. As a general point, the image quality for IHC should be improved where possible. Upon printing, the resolution of the images appears poor, making the images harder to interpret. Capturing high magnification images may help with this.

Reviewer #3 (Remarks to the Author): Expert in Cu chelators

Summary: Despite the prevalence of mutations in the small GTPase KRAS, no effective anti-RAS therapies have transitioned into the clinic and selective targeting of its downstream effectors, like MAPK components and PI3K-AKT provide limited efficacy. Therefore, many groups have focused on defining oncogenic KRAS-mediated cellular reprogramming that sustains unrestricted tumor growth, survival, and therapeutic resistance to identify novel molecularly targetable vulnerabilities.

Therefore, Aubert et al. interrogated oncogenic KRAS-driven changes in gene expression in immortalized intestinal epithelial cells and focused on significant enrichments associated with the cell surface. Towards validation of the altered transcriptional landscape driven by oncogenic KRAS, Aubert et al. employed two unbiased proteomics approaches to capture the cell surface proteome and subsequently corroborated changes in several of cell surface proteins that were altered at either or both the mRNA and protein level by immunofluorescence. Instead of creating a CRISPR library based on the transcriptomic or proteomic changes mediated by oncogenic KRAS in the intestinal epithelial cells, the authors utilized functional genomics with a previously developed surfaceome-based lentiviral CRISPR library. Intriguingly, one of the proteins that was upregulated in response to KRASG12V transformation was essential for the survival of the intestinal epithelial

cells both in vitro and in vivo expressing KRASG12V, which agrees with previously published work by Zhu et al. and Shanbhag et al. that demonstrated that ATP7A is required for RAS-driven transformation of MEFs and in in vivo cancer models of RAS activation.

In support of upregulation of the P-type ATPase ATP7A, which facilitates the transfer of Cu¹⁺ into the lumen of the secretory compartment and cellular Cu¹⁺ export when intracellular Cu levels are elevated, several other Cu-dependent protein and enzymes were upregulated in KRASG12V expressing cells and the Cu chaperone CCS that is degraded in a Cu-dependent fashion was downregulated. Further, data presented in human colon cancer cell lines and patients samples suggested that ATP7A is upregulated in a mutant KRAS selective manner and it may protect cells from excess Cu. Aubert et al. also demonstrate that the KRAS-mutant expressing normal cells or cancer cells are more sensitive to the Cu chelator tetrathiomolybdate (TTM) and this sensitivity correlates with reduced phosphorylation of ERK1/2, ceruloplasmin activity, oxidative phosphorylation, and ATP generation and suggest that the sensitivity to the Cu chelator is dependent on ATP7A. Finally, the authors explore whether Cu levels are increased in the KRAS-mutant cells via macropinocytosis and find that the macropinocytosis inhibitor EIPA increases CCS levels suggesting that Cu levels are reduced and this is important in the context of in vivo tumorigenesis.

Taken together, the paper is experimentally solid and utilizes unbiased proteomics and functional genomics approaches to identify a novel vulnerability in the context of KRAS-driven colon cancer. The findings have implications for treatment strategies in KRAS mutant colon cancer and are novel in nature. However, the main conclusion that ATP7A is upregulated in response to oncogenic KRAS-mediated macropinocytosis, which facilitates the increase in Cu uptake that is necessary for tumor cell survival, to protect the cells isn't fully addressed via the experiments presented here. Therefore, there are additional experiments that would improve the manuscript as submitted and warrant a revision.

Major Points

1. The authors suggest that intracellular Cu levels are elevated in the KRAS transformed intestinal epithelial cells, KRAS mutation-positive colon cancer cell lines, and KRAS mutant colon cancer patient samples. However, the authors never directly test Cu levels in these contexts. The authors should perform both ICP-MS for Cu levels in these samples and Cu imaging utilizing fluorescent probes to directly interrogate whether Cu levels are changed in an oncogenic KRAS-dependent fashion.
2. Similarly, the authors suggest that the intracellular Cu levels are elevated in the KRAS transformed epithelial cells and KRAS mutation-positive colon cancer cell lines because there is elevated macropinocytosis but the authors never directly test whether EIPA treatment would reduce intracellular Cu levels. The authors should perform both ICP-MS for Cu levels in these samples and Cu imaging utilizing fluorescent probes to directly interrogate whether Cu levels are changed in response to EIPA treatment.
3. It's surprising that the ATP7A knockout reduces the phosphorylation of ERK1/2 because intracellular Cu levels should be higher and thus increased Cu should be shuttled to the MEK1/2 kinases. The authors should test intracellular Cu levels in the ATP7A knockout cells and test whether previously published mechanisms contribute to the reduction of phosphorylation of ERK1/2. Namely, differential RTK (ie. EGFR) surface expression.
4. Further, the authors would predict that the KRAS-mediated increase in Cu uptake via macropinocytosis is what causes the increase in ATP7A levels to protect the cells from cuproptosis. But the cellular phenotypes of the ATP7A knockout and the Cu chelator are identical. The authors should distinguish the need for ATP7A export of Cu and the ATP7A Cu import to the secretory pathway in the KRAS-mutant cells by re-expressing mutants of ATP7A that can't traffic to the cell membrane or are preferentially targeted to the golgi. These studies will help decipher whether ATP7A plays a protective or primary role in KRAS-mediated tumorigenesis.
5. "TTM reversed the Cu-dependent increase of ATP7A levels (Extended data Fig. 6d, f), and increased CCS levels (Extended data Fig. 6f), suggesting reduced intracellular Cu levels." The

authors have not shown that the increase in ATP7A in the KRASG12V-expressing cells is due to elevated intracellular Cu levels so this should be rephrased or even better tested.

6. "Consistent with this, we found that TTM significantly reduced Cu-induced ERK1/2 phosphorylation in KRAS cells (Fig. 3e, Extended data Fig. 6f), and this decrease was found to also depend on ATP7A (Extended data Fig. 6f, g)." The authors didn't test whether Cu-induced ERK1/2 phosphorylation requires ATP7A but instead showed that either TTM or ATP7A knockout reduces ERK1/2 phosphorylation so this should be rephrased or even better tested.

Minor Points

1. The IHC should be quantified in Extended Data 5.
2. The Figure 3 legend is lacking experimental details necessary to interpret the results.

Reviewer #4 (Remarks to the Author): Expert in in vivo screens

In this study, Aubert L et al, have tried to identify Kras-specific vulnerability in colorectal cancer cells. They initially found that Kras mutation altered cell-surface protein expression and thus performed CRISPR fitness screen using surfaceome library. Through the screen, they identified that ATP7A was Kras-specific vulnerability. Mechanistically, they showed that Kras mutation upregulated macropinocytosis-mediated Cu intake, which supported tumor growth through cuproenzyme activation by ATP7A-dependent mechanism. They lastly demonstrated that blockage of ATP7A, Cu chelator or micropinocytosis could be potential therapeutic target of Kras-mutated CRC.

Overall, findings are novel and interesting besides clinically meaningful. This paper is also methodologically sound to use the brand-new technology, CRISPR screen. On the other hand, the biggest criticism is that it is not clearly shown whether ATP7A is really Kras-mutated cell-specific vulnerability. In addition, there are some questions regarding methodological details of CRISPR screen and insufficiency about the MoA especially how ATP7A mediate Copper signaling in this context. Specific questions and concerns are listed below, which need to be addressed for further consideration.

1) Basically, main theme of this paper is to identify the gene related to Kras-mutated cell-specific vulnerability. However, in the validation study, they only showed that ATP7A inhibition suppressed cell proliferation and anchorage independent growth in Kras-mutated cell lines in vitro. The effect of ATP7A inhibition on Kras-WT cells is not shown. It is also important to validate these findings in vivo because they identified ATP7A in in vivo screen, too. It is also important to use not only siRNA/shRNA but also at least 2 gRNAs identified in their screens.

2) It is unclear how increase in membrane ATP7A, which presumably mediates extracellular Cu export, contributes to intracellular ERK phosphorylation and mitochondrial respiration, when Cu uptake is increased by macropinocytosis.

3) Please check whether Kras-mutant cells have higher levels of mitochondrial respiration and ATP production than Kras-WT cells. It also needs to be addressed whether ATP7A suppression reduced levels of mitochondrial respiration and ATP production.

4) Macropinocytosis may mediate uptake of other several growth factors and cytokines. Therefore, it is important to show that in vivo anti-tumor effect of EIPA is dependent on the blockage of Cu intake but not other factors. What happens if treat Cu-deficient diet-fed xenograft mice with EIPA? Does anti-tumor effect of EIPA go away?

5) Does overexpression of ATP7A confer the resistance against Copper overload and also growth advantage in Kras-WT cells?

6) Function of ATP7A may differ between intestinal cells and other cell types. Is this Copper dependence of Kras-mutated cells general finding even in other cancer types such as PDAC?

7) Isn't it possible to directly measure the intracellular Cu levels instead of other surrogate markers such as CCS levels? It may convince us of your findings more.

8) Regarding CRISPR library method, there are several questions and concerns.

A) It is undescribed how the gene-level differential essentiality scores were calculated. Is it based on the average change of all designed gRNAs targeting same gene?

B) Number of predefined reference genes(essential and non-essential) used are fewer than the original BAGEL method. Only 193 genes, which may potentially cause bias. Assumption is Kras mutation status does not alter gene essentiality regarding these genes.

However, if it's not the case and kras mutation significantly alter dependency on many essential genes or independency on many non-essential genes, it might not be appropriate to directly compare BF calculated from BAGEL algorithms between Kras-mutated cells and Kras-wt cells. In addition, recent paper use the modified version of BAGEL to remove confident cancer genes from the reference set (Behan FM et al., Nature 2019). Please consider this point.

C) It is generally difficult to keep the library complexity in in vivo setting, which makes difficult us to perform negative CRISPR screening in vivo. Please show read count distribution in cells and tumors. How many gRNAs indeed remained inside tumors? Are most of gRNAs targeting non-essential genes still remained?

D) Contents of surfaceome-based gRNA library was not clearly written. Although it was written as gRNAs against cell-surface protein-coding genes differentially modulated by KRAS in the method, how are these genes selected? Gene list in the library seems not to match genes identified in their LC/MS-based surface protein assay.

9) It is unclear how authors chose 20 genes to show the correlation of Kras-specific changes between IEC-6 and CRC specimens in Figure 1C. It seems to be arbitrary selected. More clear explanation or comparison of global transcriptome change should be done.

10) Please perform gene annotation enrichment analysis using TCGA CRC data with or w/o Kras mutation to assess whether similar enrichment as intestinal cells is seen.

11) Description of Supplementary Table 2 is unclear. Table contains Fold-change (Kras/Control) but there is no raw number of peptide for each group. Moreover, fold changes of some genes are incomputable but log2 fold change show 15. This arbitrarily value transformation needs to be explained in the methods, if applied.

12) It is unclear what N/A means in Extended data Figure 2F.

Responses to reviewers' comments:

Reviewer #1 (Remarks to the Author): Expert in cell surface proteomics

The authors undertake a major proteogenomic study where they demonstrate that oncogenic KRASG12V expression reprograms the intestinal cell transcriptome in a manner that partially recapitulates the surfaceome (cell surface proteome).

They subsequently identify several potential therapeutic targets through cell surface-based (CRISPR/Cas9 screen using a surfaceome-based lentiviral gRNA library) loss-of-function screens. In particular, ATP7A (a copper-export protein) upregulated by mutant oncogenic KRASG12V was also upregulated at the surface of KRAS-mutated colorectal cancer cells and protects them from excess copper-ion toxicity. KRAS-mutated cells required higher Cu levels to support growth, which was found to be provided via a CTR1-independent uptake mechanism involving macropinocytosis.

The authors contend that oncogenic KRAS enhances Cu-dependent ATP7A cell-surface expression, which becomes a vulnerability and potent synthetic lethal target for KRAS-addicted CRC and that KRAS-selective vulnerability may be exploited therapeutically for treatment of KRAS-mutated cancers.

Reply: We thank the reviewer for his/her positive comments and the time spent in reviewing our manuscript.

Comments/Improvements:

1. *The manuscript is generally well written, logical, easy to follow and describes a well-constructed multi-omics approach.*

Reply: We thank the reviewer for his/her enthusiasm and the positive evaluation of our work.

2. *The manuscript would improve considerably if its strength were emphasised. This is a wonderful example of using proteogenomics (i.e., KRASG12V mediated transcriptomics with cell surface proteomics changes) to elucidate novel potential therapeutic targets through cell surface-based loss-of-function screens, in vitro and in vivo studies using Cu supplementation or chelation and comparisons with mutated CRC cell lines and CRC tissues.*

Reply: We thank the reviewer for his/her appreciation of our efforts to identify new KRAS-selective vulnerabilities using multi-omics approaches. As suggested, we have modified the manuscript to better highlight its different strengths. For example, we now state the advantages of using two chemoproteomic approaches, as well as the complementarity of the transcriptome and surface proteome induced by oncogenic KRAS.

3. *Technical levels of stringency employed to quantitatively measure specific mRNA and protein level changes in the KRASG12V-mediated transcriptome and cell surface proteome must be explained/enumerated in much greater detail. For example, a study of this importance would be expected to rely on high-stringency proteomics data - whereby the number of peptides used should be 2 or more, the length of non-nested peptides used should be 9 amino acids or more, FDRs should be <1% for both the peptide and protein FDRs, leucine/isoleucine switching*

should be checked along with use of the neXtProt peptide untypicity checker to ensure peptides used for a protein call are unique-expressed for any given protein.

Reply: We thank the reviewer for these important comments. We have modified the Methods section to explain how RNA-seq data were processed, and also described the different measures used to generate high-stringency proteomics data. With regards to the latter, the manuscript now indicates that, during data collection, the false discovery rate (FDR) for peptide and protein was set to 0.5%. The minimum number of peptides per protein was set to 2, and minimum peptide length was set to ~6 amino acids. Search criteria included a static modification of cysteine residues of +57.0214 Da; a variable modification of +15.9949 Da to include potential oxidation of methionine; and a modification of +79.966 on serine, threonine, or tyrosine for the identification of phosphorylation. Proteins were selected based on their detection in at least two replicates of the same biological condition, with a minimum of 2 unique peptides, which were verified for uniqueness and leucine/isoleucine switch using the neXtProt checker (PMID: 28520855). We hope that these precisions will satisfy the reviewer about the quality of our proteomics and transcriptomics data.

4. *Although their data from CSB and CSC chemoproteomics approaches compliment each other, the reason why both were employed in the “study” is not self-evident - this could be clarified.*

Reply: We have modified the manuscript to emphasize the complementarity of the chemoproteomic approaches used to identify KRAS-dependent changes at the cell surface. Briefly, CSB and CSC employ different biotin reagents to label and isolate cell surface proteins (PMID: 18763706). While CSB (which uses sulfo-NHS-LC-biotin) relies on the labeling free primary amines in cell-surface proteins, CSC (which uses aminooxybiotin) relies on the labeling of cell-surface sialylated glycoproteins. During the optimization phase of the project, we realized that CSB and CSC differed in their capacity to enrich in particular cell-surface proteins (see Figure 1g and Extended data Figure 2c), likely due to differences in the accessibility of functional groups and/or in the labeling of particular types of cell-surface proteins. Thus, we chose to use both CSB and CSC in parallel to maximize our capacity to globally identify/quantify cell-surface proteins. This is in fact a strength of the study, and we have made sure to emphasize this in the manuscript.

5. *Again, a rationale for why label-free quantification by liquid chromatography coupled with tandem mass spectrometry (LC-MS/MS) and not TMT-labelling was employed would have been helpful.*

Reply: This is again an excellent point that we have actually addressed experimentally during the optimisation phase of the project. Notwithstanding the fact that some biotin reagents interfere with TMT labeling, we tested whether this approach would improve the quantification of peptides derived from cell-surface proteins. After a few attempts, it became clear that the addition of several purification steps required for TMT labeling drastically reduced the yield of peptides for MS/MS analysis. Due to this issue, we chose to optimize our surfaceome protocols with label-free quantitative proteomics and ensured that the level of reproducibility for cell surfaceome changes was reliable enough, reaching similar reproducibility rates (Extended data Figure 2d) as observed by other groups (PMID: 26699813; 27525720). We also validated many hits with at least three

different methods (IF, WB, Flow cytometry), as demonstrated in Extended data Figure 3. We hope this answers the question raised by the reviewer.

6. *The involvement of ATP7A in mutant KRAS cell fitness was confirmed by RNAi using IEC-6 cells and the CRC cell lines HCT116 and DLD-1 cells. Whilst the mutational variation between lines HCT116 and DLD-1 cells is well known, the reason for initial choice of the IEC-6 model as the most appropriate could be further explained.*

Reply: The main reason for using IEC-6 cells is that they are derived from intestinal epithelial crypt cells and considered normal and non-cancerous (PMID: 88453). The introduction of oncogenic KRAS was shown to fully transform these cells (PMID: 25500543; 21699772), which provided an isogenic system for specifically studying the effect of KRAS-mediated transformation. CRC cell lines harbor heterogeneous sets of mutations, which would have increased the complexity in assigning particular cell-surface changes to KRAS mutations.

7. *Equally, to verify oncogenic KRAS augments Cu levels in human CRC cells, they evaluated CCS and ATP7A levels in KRAS wild-type (CACO-2) and KRAS-mutated (HCT116, DLD-1, SW480, SW620) cells and observed that KRAS-mutated cells had low CCS levels inversely correlated with high ATP7A levels. Explain why they chose to use so many different CRC cell lines here?*

Reply: As indicated above, we chose to verify our findings in a larger set of CRC cell lines, as they are quite heterogeneous in nature. Nevertheless, we were able to validate our results to reasonably conclude that KRAS mutation is associated with low CCS and high ATP7A levels. We hope this explains our choice of using several CRC cell lines.

8. *The effects of using a Cu-supplement diet and chelators are convincing. Did they consider using proteogenomics to determine off-target effects that may have been induced by these treatments?*

Reply: We thank the reviewer for this interesting suggestion, but we have not performed transcriptomic or proteomic screens using Cu-supplement diet or chelators. While off-target effects due to Cu supplementation are difficult to imagine, chelators could possibly have off-target effects by blocking other important metals. However, tetrathiomolybdate (TTM) is considered to be remarkably selective for Cu, as TTM-mediated Cu deprivation can usually be reversed by Cu supplementation *in vitro* and *in vivo* (PMID: 7264629). For these reasons, we think that the risk of observing off-target effects with Cu or TTM is relatively low. In the context of another study, we have performed CRISPR/Cas9-based chemogenomic screens using TTM, which revealed *ATP7A* and *ATOX1* as the highest scoring rescue genes. The scores for the two Cu genes were significantly higher than any other genes, indicating that TTM mainly acts by antagonizing Cu metabolism. Regarding the Cu-supplement diet, we followed previously established *in vivo* protocols (PMID: 24717435) and validated Cu deficiency using the well established clinical marker Ceruloplasmin activity (PMID: 18598583; 19916391), which was reduced due to Cu deficiency. Nonetheless, we acknowledge that due to the wide role played by Cu in cells, several important cellular processes may be affected by Cu deficiency, but we do not consider these effects as off-target.

Reviewer #2 (Remarks to the Author): Expert in macropinocytosis and ras-driven cancer

Review of manuscript “Copper bioavailability is a KRAS-specific vulnerability in colorectal cancers”

In the submitted study, Aubert et al. provide evidence that overexpression of mutant KRAS is sufficient to modulate the expression of many cell-surface proteins, both at an mRNA (the transcriptome) and protein (surfaceome) level. Using a CRISPR/Cas9 loss-of-function screen, the authors identify cell-surface proteins that contribute to the fitness of mutant KRAS cells. From here, the work shifts to focus on one of their most promising hits, ATP7A, a copper exporter. Using protein levels as a biomarker of intracellular copper concentrations, the authors go on to show that mutant-Ras colorectal cell lines are more sensitive to copper chelation, and that indeed, levels of both ATP7A and CCS, copper chaperone for superoxide dismutase, vary with mutant-Ras status in a way that links Ras activity to increased intracellular copper levels. Upon increases in intracellular Cu, ATP7A will be upregulated to ensure sufficient export to prevent Cu toxicity. CCS in turn will be degraded in the presence of high Cu concentrations. Somewhat paradoxically, however, the authors also show that CTR1, the primary copper importer in cells, is actually downregulated, which they state implies that Cu must be entering the cell by other means in the Ras mutant setting. The authors follow up this in vitro and in silico work with an in vivo loss-of-function screen to validate their findings and conclude by showing that macropinocytosis may facilitate copper uptake into Ras-mutant colorectal cancer cells in lieu of CTR1 levels being downregulated, contributing to their overall fitness. Inhibiting macropinocytosis by treating tumors with EIPA can prevent the rescue of tumor growth by CuSO4 supplementation in the Cu-deficient setting.

The paper looks to be the first to undertake a full loss-of-function screen (both in vitro and in vivo) to identify Ras-dependent changes to cell surface proteins, whereas previous attempts have focused on mutational analysis (PMID: 25193853). Taken as a whole, the submitted work offers a promising glimpse into what may be a novel metabolic vulnerability in Ras-mutant colorectal cancer mediated by cell surface ATP7A, which would be of interest to both the Ras biology and oncology fields. While there is some work to indicate that copper is critical in certain cancers subtypes (PMID: 30824611), this is the first to link this vulnerability directly to mutant Ras and such an advance would be a welcome addition to the aforementioned fields. However, multiple of the authors' claims are left under supported, primarily in the second half of the paper in which a dissection of the importance of copper import is argued, but intracellular copper concentrations are never directly measured.

Finally, the argument that “macropinocytosis is required for copper entry into KRAS tumors” is left severely underdeveloped and in need of significantly more data and critical control experiments to support this conclusion. Specific comments are now listed below.

Reply: We thank the reviewer for his/her very encouraging comments, constructive criticisms and accurate description of our findings. We also thank him/her for taking the time to review the manuscript. In particular, we appreciate the recognition that our paper highlighted a novel druggable metabolic vulnerability in Ras-mutant colorectal cancer.

Major Comments:

1. *The authors argue that ATP7A is a potent synthetic lethal target for KRAS mutant CRC cells. To prove this claim, the authors would be required to show that ATP7A knockdown in the context of non-Ras mutant lines has little to no effect, whereas the lethality in Ras mutant cells will be very high. We note that the target itself is derived from a synthetic lethal CRISPR screen, but nonetheless this hit should be validated in a thorough way. While the data in Extended data Figure 4e and 4f show that siRNA targeting of ATP7A limits cell doubling in Ras-mutant cells, they do not provide any evidence that this effect is not also seen in Ras-WT cells (as they have for copper supplementation in Figure 2g).*

Reply: We thank the reviewer for this comment, which we addressed using RNAi in both IEC-6 cells and CRC cell lines. As shown in Figure 2e, we found that KRAS IEC-6 cells were significantly more affected than wild-type counterparts by ATP7A depletion using two different target sequences. These results are consistent with our CRISPR/Cas9 screen (Figure 2a-d), which found that 7 out of 8 gRNAs targeting ATP7A significantly depleted KRAS IEC-6 cells, compared to wild-type counterparts (Figure 2d, Extended data Figure 4a). Similar results were obtained with KRAS-mutated (HCT116 and SW620) CRC cell lines, which were found to be more sensitive than CACO-2 cells (KRAS-wt) to ATP7A depletion (Figure 2f). Together, we believe that our data convincingly show that ATP7A is a potent synthetic lethal target for KRAS-mutated cells.

2. *While the entire paper revolves around the premise that high intracellular Cu levels indicate a novel metabolic vulnerability in Ras-mutant cells, the authors only use indirect measures (albeit well established in the current literature) of Cu levels in the paper. Not one experiment measures the concentration of copper itself. This is especially glaring in the final figure of the main body of the text, in which the authors propose that macropinocytosis is a key import mechanism for Cu in Ras-mutant cells. To argue this claim, data would be required showing definitively that copper levels drop to extremely low levels in the presence of a macropinocytosis inhibitor.*

Reply: We thank the reviewer for this comment, and we certainly agree with the need for alternative approaches to measure Cu levels in cells. In this context, it is important to note that dependent on the kinetic accessibility, cellular Cu can be divided into a canonical, static pool (i.e., Cu bound to enzymes such as cytochrome c oxidase) and an exchangeable, labile pool (i.e., Cu bound to chaperones such as CCS). The latter form of Cu is more bioavailable and can participate in dynamic cell signaling pathways (PMID: 25692243). As indicated by the reviewer, CCS levels are a well-established readout that correlates with bioavailable Cu levels (PMID: 16950140; 28931909). Our results show that KRAS-mutated cells have decreased CCS levels compared to Control counterparts (Figure 3c), suggesting that oncogenic KRAS increases Cu bioavailability in cells. We also show that inhibition of macropinocytosis increases CCS levels in KRAS-mutated cells but not in Control cells (Figure 5c), suggesting that macropinocytosis is an important entry route for bioavailable Cu in KRAS-mutated cells.

There are alternative technologies to measure intracellular Cu levels, such as inductively coupled plasma-mass spectrometry (ICP-MS), but these types of approaches detect total metal content,

which does not necessarily reflect the biologically active or bioavailable Cu pools (PMID: 16950140; 22900086; 28617866; 28931909). For example, it was shown that CTR1 knockout leads to Cu accumulation and a paradoxical increase in CCS levels, suggesting that enhanced Cu levels does not necessarily correlate with its bioavailability and capacity to regulate the activity of cuproenzymes (PMID: 16950140). Nonetheless, we used ICP-MS to analyze total Cu content, and found that Cu levels were higher in KRAS-mutated CRC cells compared to wild-type counterparts (Extended data Figure 5I).

As an alternative to CCS levels as a measure of bioavailable Cu, we decided to use Cu-sensing fluorescent probes, which provide relative information on the bioavailability and distribution of labile Cu pools in cells (PMID: 25692243). For this, we initiated a collaboration with Dr. Christoph Fahrni (Georgia Institute of Technology), who developed a ratiometric Cu(I)-selective fluorescent probe (crisp-17) that was recently used to probe labile Cu pools in ATP7A-mutated patient fibroblasts (PMID: 31160463). Using this probe, we observed that KRAS-mutant IEC-6 and CRC cells responded to DTDP treatment, an oxidant that is shown to release thiol-bound cellular Cu (Figure 3d, g). Notably, the kinetics for gradual regression of the initial ratio increase upon DTDP addition was significantly slower for KRAS cells relative to their wild-type counterparts, suggesting an increased depth of the bioavailable thiol-bound Cu pool in KRAS mutant cells. We also observed that treatment of KRAS-mutated cells with inhibitors of macropinocytosis exhibited a small but significant decrease in fluorescence ratio, suggesting reduced bioavailable Cu levels (Figure 5d). Consistent with these results, CCS levels were increased upon inhibition of macropinocytosis (Figure 5c). Finally, using ICP-MS we also observed that the levels of total Cu are higher in KRAS-mutated CRC compared to wild-type counterparts (Extended data Figure 5I). Altogether, the additional ratiometric imaging and ICP-MS data are consistent with an increase in Cu bioavailability in oncogenic KRAS, and that macropinocytosis is an important entry route.

3. *The in-vivo macropinocytosis experiment (Figure 4e) excludes a vital control. Since, as has been previously established in published work (PMID: 23665962) and as the authors acknowledge directly in Figure 4b, Ras-mutant cells are selectively sensitive to inhibition of macropinocytosis by EIPA when compared to non-mutant controls. This necessitates that for their in vivo experiment they add at least one more arm containing EIPA treatment alone. The effects seen from the CuSO₄ to the CuSO₄ + EIPA arm may have little to do with preventing copper uptake and may actually occur due to the deleterious effects of EIPA independent of this (likely amino acid deprivation). Also, this figure offers little insight into the selectivity of these effects for Ras-mutant cells in vivo, as only a mutant-Ras line is tested.*

Reply: We thank the reviewer for pointing out this experimental control, which was actually part of our initial experimental plan. Unfortunately, the combination of Cu deficiency and EIPA treatment resulted in the loss of all animals that were part of this experimental paradigm. These mice rapidly reached critical endpoints due to weight loss, visible dehydration (EIPA is a diuretic), and general signs of distress, which led to their prompt euthanasia. These endpoints were carefully chosen based on what is expected from prolonged Cu deficiency and EIPA treatment (PMID: 29959467; 9782174) and were approved by the Université de Montréal Animal Care Committee, which applies guidelines set by the Canadian Council on Animal Care (CCAC). Notwithstanding

the likely opposition that we would face from the Animal Care Committee if we were to propose to repeat this control condition, we feel that it would be ethically questionable to do so.

That said, we do not feel that this control is required to conclude that macropinocytosis is involved in Cu uptake by KRAS-mutant cells *in vivo*. We have been very careful not to over interpret our findings, by stating that “...blocking macropinocytosis reduces Cu uptake, Cu-dependent signaling, as well as tumor growth”. In fact, we argue that many nutrients, in addition to Cu, are likely to enter KRAS-mutated cells *via* macropinocytosis. Therefore, we expect that EIPA treatment would also have Cu-independent effects on tumor growth, as was shown by its involvement in amino acid uptake (PMID: 23665962). The multiple roles of macropinocytosis in tumor growth make it difficult - if not impossible - to identify a single cause for the decrease in tumor growth observed in response to EIPA treatment.

Minor comments:

1. *In Figure 1c, it is unclear how the gene set was generated. In the text, they write “[...]we found a very good correlation between our dataset and oncogenic KRAS-specific changes occurring in CRC specimens from The Cancer Genome Atlas[...],” however it not clear if these are simply all the genes differentially regulated by KRAS that are common between the data set or the most statistically significant hit or if this data set was created by some other criteria. Some clarity here would be helpful for the reader.*

Reply: We apologize for the lack of clarity on how we designed the heatmap. The methods section and our conclusions in the main body of the manuscript have been adapted to provide more clarity. Briefly, using cBioPortal (<https://www.cbioportal.org/>), we first isolated the list of genes most differentially expressed in KRAS-altered compared to KRAS-unaltered CRC specimens (TCGA, 594 samples). We then compared this list to the genes differentially expressed in KRAS-mutated IEC-6 cells, compared to control counterparts, and derived common genes between the two datasets. We also filtered the dataset to identify genes that were previously reported to be regulated by oncogenic KRAS. We then selected 20 genes (10 over- and 10 under-expressed) to generate the heatmap shown in Figure 1c.

2. *Figure 2c, and 2e; extended data figure 3c (TLR4 graph); extended data 4a, 4b (surfaceome bar), 4e, and 4f; extended data figure 6a, and 6i; and extended data figure 7d are missing errors bars..*

Reply: We thank the reviewer for this comment. We added missing error bars to all graphs, except those presenting Bayes Factor (BF), which is a measure of confidence taking into account the value of all replicates.

3. *For consistency, all single color images should be presented in black and white, as the authors have done in extended data figure 3b.*

Reply: As suggested, we have homogenized all immunofluorescence microscopy images to be consistent across them all.

4. *In extended data Figure 4i, j, the authors present cell death fold change, whereas in Figure 2g, i they present % cell number, and in extended data figure 6b, c, and e, they present transformed growth %. Readouts for cell fitness should be consistent throughout.*

Reply: We thank the reviewer for this comment, as it shows that we need to be clearer with the assays used in the study. The main reason for the different data presentation is that we do not interpret the readouts of the different assays in the same way. The Extended data Figures 4i and 4j (= new Extended data Figs. 5k and 5m) measure dead cells stained by Annexin-V and are hence displayed as relative fold-change in cell death. For Figures 2g and 2i (= new Figs. 3e and 3h), since we only counted live cells, it is an indication of proliferation and anchorage-dependent growth, and therefore presented as percentage of live cells. For soft agar assays presented in Extended data Figures 6b, 6c and 6e (= new Extended data Figs. 4f, 4g and 7g), the data measures anchorage-independent growth and are therefore presented as percentage of transformed growth, which is typical for these kind of data. All these readouts are complementary for monitoring cell fitness and were performed in KRAS-transformed IEC-6 *versus* control IEC-6 (when possible), and in at least two KRAS-mutated *versus* KRAS-wt CRC cell lines, thereby increasing the relevance of our findings.

5. *In Figure 3e bar graphs, what precise treatment group is being quantified?*

Reply: For Figure 3e (= new Fig. 4d), we acknowledge that the annotations are not obvious. The bars represent non-treated cell lysates (black) and TTM-treated cell lysates (red). We made the appropriate changes, as now presented in Figure 4d.

6. *In extended data figure 4h, authors should include the control cell images and specify which images are control and which are ras-mutant cells.*

Reply: As suggested, we have added the Control IEC-6 images with/without treatment along with the KRAS mutant cells to complete the figure (= new Extended data Fig. 5g-j), including the profiles that demonstrate the intracellular distribution of ATP7A between the conditions.

7. *Figure 4c bar graph should be consistent with color scheme throughout the whole paper. Vehicle bar should be colored blue or black, as authors have done throughout.*

Reply: We have adapted Figure 4c (= new Fig. 5c) to be consistent with the color scheme used throughout our study.

8. *Quantification of IHC data is artificially overpowered by including each counted field as an individual data point. The statistical power from such a study is derived from the number of mice used, not the number of images analyzed. Here, n values should be the number of mice on each treatment arm. We recommend that the authors take the average quantification of the images from each mouse and provide that as one value, deriving the error instead from the values of each mouse, leaving n = 6. While this will make achieving statistical significance more difficult, it is a more accurate way to present the data since it is unclear if the values driving results are all derived from outlier mice.*

Reply: As suggested, we re-analyzed the IHC data to represent the data as one value per mouse. As shown in the graphs below, this modification did not significantly alter the results and our conclusions based on these data. *P*-values were measured using a two-tailed, unpaired Student's *t*-test.

9. As a general point, the image quality for IHC should be improved where possible. Upon printing, the resolution of the images appears poor, making the images harder to interpret. Capturing high magnification images may help with this.

Reply: We adjusted the quality of all IHC images for the final version of the manuscript.

Reviewer #3 (Remarks to the Author): Expert in Cu chelators

Summary: Despite the prevalence of mutations in the small GTPase KRAS, no effective anti-RAS therapies have transitioned into the clinic and selective targeting of its downstream effectors, like MAPK components and PI3K-AKT provide limited efficacy. Therefore, many groups have focused on defining oncogenic KRAS-mediated cellular reprogramming that sustains unrestricted tumor

growth, survival, and therapeutic resistance to identify novel molecularly targetable vulnerabilities.

Therefore, Aubert et al. interrogated oncogenic KRAS-driven changes in gene expression in immortalized intestinal epithelial cells and focused on significant enrichments associated with the cell surface. Towards validation of the altered transcriptional landscape driven by oncogenic KRAS, Aubert et al. employed two unbiased proteomics approaches to capture the cell surface proteome and subsequently corroborated changes in several of cell surface proteins that were altered at either or both the mRNA and protein level by immunofluorescence. Instead of creating a CRISPR library based on the transcriptomic or proteomic changes mediated by oncogenic KRAS in the intestinal epithelial cells, the authors utilized functional genomics with a previously developed surfaceome-based lentiviral CRISPR library. Intriguingly, one of the proteins that was upregulated in response to KRASG12V transformation was essential for the survival of the intestinal epithelial cells both in vitro and in vivo expressing KRASG12V, which agrees with previously published work by Zhu et al. and Shanbhag et al. that demonstrated that ATP7A is required for RAS-driven transformation of MEFs and in in vivo cancer models of RAS activation.

In support of upregulation of the P-type ATPase ATP7A, which facilitates the transfer of CuI+ into the lumen of the secretory compartment and cellular CuI+ export when intracellular Cu levels are elevated, several other Cu-dependent protein and enzymes were upregulated in KRASG12V expressing cells and the Cu chaperone CCS that is degraded in a Cu-dependent fashion was downregulated. Further, data presented in human colon cancer cell lines and patients samples suggested that ATP7A is upregulated in a mutant KRAS selective manner and it may protect cells from excess Cu. Aubert et al. also demonstrate that the KRAS-mutant expressing normal cells or cancer cells are more sensitive to the Cu chelator tetrathiomolybdate (TTM) and this sensitivity correlates with reduced phosphorylation of ERK1/2, ceruloplasmin activity, oxidative phosphorylation, and ATP generation and suggest that the sensitivity to the Cu chelator is dependent on ATP7A. Finally, the authors explore whether Cu levels are increased in the KRAS-mutant cells via macropinocytosis and find that the macropinocytosis inhibitor EIPA increases CCS levels suggesting that Cu levels are reduced and this is important in the context of in vivo tumorigenesis.

Taken together, the paper is experimentally solid and utilizes unbiased proteomics and functional genomics approaches to identify a novel vulnerability in the context of KRAS-driven colon cancer. The findings have implications for treatment strategies in KRAS mutant colon cancer and are novel in nature. However, the main conclusion that ATP7A is upregulated in response to oncogenic KRAS-mediated macropinocytosis, which facilitates the increase in Cu uptake that is necessary for tumor cell survival, to protect the cells isn't fully addressed via the experiments presented here. Therefore, there are additional experiments that would improve the manuscript as submitted and warrant a revision.

Reply: We thank the reviewer for thoroughly summarizing our work, and for the positive and constructive criticisms. We are also pleased that the reviewer judges that our work is experimentally solid and novel in nature. However, we would like to clarify that we have not used a previously developed lentiviral CRISPR library. In fact, we have developed a new gRNA library

based on our surface proteomic results, by including gRNAs against all genes encoding surface proteins that were either down- or up-regulated by oncogenic KRAS. Also, we would like to clarify that, although our results suggest that macropinocytosis is involved in Cu uptake, it may not be the only mechanism by which KRAS-mutant cells increase intracellular Cu.

Major Points

1. *The authors suggest that intracellular Cu levels are elevated in the KRAS transformed intestinal epithelial cells, KRAS mutation-positive colon cancer cell lines, and KRAS mutant colon cancer patient samples. However, the authors never directly test Cu levels in these contexts. The authors should perform both ICP-MS for Cu levels in these samples and Cu imaging utilizing fluorescent probes to directly interrogate whether Cu levels are changed in an oncogenic KRAS-dependent fashion.*
2. *Similarly, the authors suggest that the intracellular Cu levels are elevated in the KRAS transformed epithelial cells and KRAS mutation-positive colon cancer cell lines because there is elevated macropinocytosis but the authors never directly test whether EIPA treatment would reduce intracellular Cu levels. The authors should perform both ICP-MS for Cu levels in these samples and Cu imaging utilizing fluorescent probes to directly interrogate whether Cu levels are changed in response to EIPA treatment.*

Reply: We thank the reviewer for these related comments, and we certainly agree with the need for alternative approaches to measure Cu levels in cells. In this context, it is important to note that depending on the kinetic accessibility, cellular Cu can be divided into a canonical, static pool (i.e., Cu bound to enzymes such as cytochrome c oxidase) and an exchangeable, labile pool (i.e., Cu bound to chaperones such as CCS). The latter form of Cu is more bioavailable and can participate in dynamic cell signaling pathways (PMID: 25692243; 25692243). As indicated by reviewer 2, CCS levels are a well-established readout that correlates with bioavailable Cu levels (PMID: 16950140; 28931909). Our results show that KRAS-mutated cells have decreased CCS levels compared to control counterparts (Figure 3c), suggesting that oncogenic KRAS increases Cu bioavailability in cells. We also show that inhibition of macropinocytosis increases CCS levels in KRAS-mutated cells, but not in control cells (Figure 5c), suggesting that macropinocytosis is an important entry route for Cu that becomes bioavailable in KRAS-mutated cells.

As indicated by the reviewer, there are alternative technologies to measure intracellular Cu levels, such as inductively coupled plasma-mass spectrometry (ICP-MS), but these approaches detect total metal content, which does not necessarily reflect the biologically active or bioavailable Cu pools (PMID: 16950140; 22900086; 28617866; 28931909). For example, it was shown that CTR1 knockout leads to Cu accumulation and a paradoxical increase in CCS levels, suggesting that enhanced Cu levels do not necessarily correlate with Cu bioavailability and/or capacity to regulate the activity of cuproenzymes (PMID: 16950140). As an alternative to CCS levels, we decided to use Cu-sensing fluorescent probes, which provide relative information on the bioavailability and distribution of labile Cu pools in cells (PMID: 25692243). For this, we initiated a collaboration with Dr. Christoph Fahrni (Georgia Institute of Technology), who developed a ratiometric Cu(I)-selective fluorescent probe (crisp-17) that was recently used to probe labile Cu pools in ATP7A-mutated patient fibroblasts (PMID: 31160463). Using this probe, we observed that KRAS-mutant

IEC-6 and CRC cells responded to DTDP treatment, an oxidant that is shown to release thiol-bound cellular Cu (Figure 3d,g). Notably, the kinetics for gradual regression of the initial ratio increase upon DTDP addition was significantly slower for KRAS cells relative to their wild-type counterparts, suggesting an increased depth of the bioavailable thiol-bound Cu pool in KRAS mutant cells. We also observed that treatment of KRAS-mutated cells with inhibitors of macropinocytosis exhibited a small but significant decrease in fluorescence ratio, suggesting reduced bioavailable Cu levels (Figure 5d). Consistent with these results, the CCS levels were increased upon inhibition of macropinocytosis (Figure 5c).

Finally, using ICP-MS we also observed that the levels of total Cu are higher in KRAS-mutated CRC compared to wild-type counterparts (Extended data Figure 5l). Altogether, the additional ratiometric imaging and ICP-MS data are consistent with an increase in Cu bioavailability in oncogenic KRAS, and that macropinocytosis is an important entry route, thus reflecting our main conclusions.

- 3. It's surprising that the ATP7A knockout reduces the phosphorylation of ERK1/2 because intracellular Cu levels should be higher and thus increased Cu should be shuttled to the MEK1/2 kinases. The authors should test intracellular Cu levels in the ATP7A knockout cells and test whether previously published mechanisms contribute to the reduction of phosphorylation of ERK1/2. Namely, differential RTK (ie. EGFR) surface expression.*

Reply: We thank the reviewer for this interesting suggestion, but it is important to note that ATP7A has a bi-functional role in cells: a homeostatic role by exporting Cu outside the cell, and a biosynthetic role by promoting the metallation and maturation of multiple cuproenzymes (PMID: 17615395, 20454597). While we expect that ATP7A knockdown would increase intracellular Cu content (PMID: 241745620), the literature also shows that this increase does not necessarily correlate with increased Cu bioavailability (PMID: 22900086; 16950140; 28931909), as measured using CCS levels and/or by testing the activity of Cu-dependent enzymes (i.e., MEK1/2 for ERK1/2 phosphorylation). Furthermore, several groups have reported that ATP7A knockdown reduces the activity of cuproenzymes, such as tyrosinase (PMID: 18650808), SSAO (PMID: 30199530) and more recently, LOXL (PMID: 30890638). Consistent with these findings, we observed that ATP7A knockdown increased CCS levels in KRAS IEC-6 cells, which is suggestive of reduced Cu bioavailability (Figure 4a). Also, we found that ATP7A knockdown resulted in reduced ceruloplasmin activity (Extended data Figure 7a), ERK1/2 phosphorylation (Extended data Figure 7b) and cytochrome C oxidase activity (Extended data Figure 7h). As another example for the biosynthetic role of ATP7A, we found that Cu chelation by TTM, which increases CCS levels and therefore diminishes Cu bioavailability, is accompanied by decreased ATP7A levels and reduced ERK1/2 phosphorylation and mitochondrial respiration (Figures 4b, d, g and h). Altogether, our data indicate that ATP7A knockdown or Cu chelation reduces bioavailable Cu levels, which decreases the activity of several Cu-dependent enzymes.

- 4. Further, the authors would predict that the KRAS-mediated increase in Cu uptake via macropinocytosis is what causes the increase in ATP7A levels to protect the cells from cuproptosis. But the cellular phenotypes of the ATP7A knockout and the Cu chelator are identical. The authors should distinguish the need for ATP7A export of Cu and the ATP7A Cu*

import to the secretory pathway in the KRAS-mutant cells by re-expressing mutants of ATP7A that can't traffic to the cell membrane or are preferentially targeted to the golgi. These studies will help decipher whether ATP7A plays a protective or primary role in KRAS-mediated tumorigenesis.

Reply: While it would be interesting to determine the respective contribution of the homeostatic and biosynthetic roles of ATP7A in KRAS-mediated tumorigenesis, distinguishing between the two is a tremendously daunting task, as it would involve a knockdown/knockout-rescue approach for a protein (ATP7A) that we found to be essential for KRAS-mutated cells. Nevertheless, we have contacted Dr. Van de Sluis (Groningen University), from whom we obtained ATP7A mutants that are either excluded from the Golgi or that cannot traffic to the cell surface (PMID: 21667063). Dr. Van de Sluis warned us that the plasmids are very difficult to propagate in bacteria, which is a well-documented problem that appears to be due to the instability of the ATP7A cDNA (PMID: 27337370). We tried different approaches to increase plasmid stability, but we could not successfully propagate plasmids to be used for subcloning ATP7A in retroviral vectors. Because we could not complete these experiments, and our lab is now closed for an unknown number of weeks/months, we made sure that our conclusions are in line with our results.

5. *“TTM reversed the Cu-dependent increase of ATP7A levels (Extended data Fig. 6d, f), and increased CCS levels (Extended data Fig. 6f), suggesting reduced intracellular Cu levels.” The authors have not shown that the increase in ATP7A in the KRASG12V-expressing cells is due to elevated intracellular Cu levels so this should be rephrased or even better tested.*

Reply: While ATP7A levels are known to be regulated by Cu bioavailability (PMID: 23174565; PMID: 28931909), we agree with the reviewer that our data do not specifically show that oncogenic KRAS increases ATP7A levels in a Cu-dependent manner. As suggested, we modified the manuscript to better reflect this fact.

6. *“Consistent with this, we found that TTM significantly reduced Cu-induced ERK1/2 phosphorylation in KRAS cells (Fig. 3e, Extended data Fig. 6f), and this decrease was found to also depend on ATP7A (Extended data Fig. 6f, g).” The authors didn't test whether Cu-induced ERK1/2 phosphorylation requires ATP7A but instead showed that either TTM or ATP7A knockout reduces ERK1/2 phosphorylation so this should be rephrased or even better tested.*

Reply: Again, we agree with the reviewer and rephrased our conclusions accordingly.

Minor Points

1. *The IHC should be quantified in Extended Data 5.*

Reply: The IHC images presented in Extended data Figure 5 (= new Extended data Fig. 6) have already been quantified and shown in Figures 3i and 3j. With all these data, we understand that it is a quite challenge to keep track of all graphs and quantifications. We thus thank the reviewer for his/her time in going through our manuscript.

2. The Figure 3 legend is lacking experimental details necessary to interpret the results.

Reply: We added more information to the legend of Figure 3 (= new Fig. 4) to facilitate the interpretation of the results.

Reviewer #4 (Remarks to the Author): Expert in *in vivo* screens

In this study, Aubert L et al, have tried to identify Kras-specific vulnerability in colorectal cancer cells. They initially found that Kras mutation altered cell-surface protein expression and thus performed CRISPR fitness screen using surfaceome library. Through the screen, they identified that ATP7A was Kras-specific vulnerability. Mechanistically, they showed that Kras mutation upregulated macropinocytosis-mediated Cu intake, which supported tumor growth through cuproenzyme activation by ATP7A-dependent mechanism. They lastly demonstrated that blockage of ATP7A, Cu chelator or macropinocytosis could be potential therapeutic target of Kras-mutated CRC.

Overall, findings are novel and interesting besides clinically meaningful. This paper is also methodologically sound to use the brand-new technology, CRISPR screen. On the other hand, the biggest criticism is that it is not clearly shown whether ATP7A is really Kras-mutated cell-specific vulnerability. In addition, there are some questions regarding methodological details of CRISPR screen and insufficiency about the MoA especially how ATP7A mediate Copper signaling in this context. Specific questions and concerns are listed below, which need to be addressed for further consideration.

Reply: We thank the reviewer for his/her positive comments and the time spent in reviewing our manuscript. We also thank the reviewer for the positive comments and constructive criticisms. In particular, we appreciate the recognition that our findings are novel and interesting, besides being clinically meaningful, and that our paper is methodologically sound.

1) Basically, main theme of this paper is to identify the gene related to Kras-mutated cell-specific vulnerability. However, in the validation study, they only showed that ATP7A inhibition suppressed cell proliferation and anchorage independent growth in Kras-mutated cell lines in vitro. The effect of ATP7A inhibition on Kras-WT cells is not shown. It is also important to validate these findings in vivo because they identified ATP7A in in vivo screen, too. It is also important to use not only siRNA/shRNA but also at least 2 gRNAs identified in their screens.

Reply: We thank the reviewer for this comment, which we addressed using RNAi in both IEC-6 cells and CRC cell lines. As shown in Figure 2e, we found that KRAS IEC-6 cells were significantly more affected than their wild-type counterparts by ATP7A depletion using two different target sequences. These results are consistent with our CRISPR/Cas9 screen (Figure 2a-d), which found that 7 out of 8 gRNAs targeting ATP7A significantly depleted KRAS IEC-6 cells, compared to wild-type counterparts (Figure 2d, Extended data Figure 4a). Similar results were obtained with KRAS-mutated (HCT116 and SW620) CRC cell lines, which were found to be more sensitive than CACO-2 cells (KRAS-wt) to ATP7A depletion (Figure 2f). Together, we believe

that our data convincingly show that ATP7A is a potent synthetic lethal target for KRAS-mutated cells.

2) *It is unclear how increase in membrane ATP7A, which presumably mediates extracellular Cu export, contributes to intracellular ERK phosphorylation and mitochondrial respiration, when Cu uptake is increased by macropinocytosis.*

Reply: In response to this comment, it is important to note that ATP7A has a bi-functional role in cells: a homeostatic role by exporting Cu outside the cell, and a biosynthetic role inside the cell by promoting the metallation and maturation of multiple cuproenzymes in the trans-Golgi network (TGN) (PMID: 17615395, 20454597). We found that oncogenic KRAS increases ATP7A expression at the plasma membrane and the TGN (Figures 3a, b, i), suggesting that both functions of ATP7A are increased in KRAS-mutated cells. These results are consistent with studies showing that ATP7A knockdown reduces the activity of several cuproenzymes, including tyrosinase (PMID: 18650808), SSAO (PMID: 30199530), and LOXL (PMID: 30890638). We hope that this helps clarify the potential role of ATP7A in KRAS-mutant cells.

3) *Please check whether Kras-mutant cells have higher levels of mitochondrial respiration and ATP production than Kras-WT cells. It also needs to be addressed whether ATP7A suppression reduced levels of mitochondrial respiration and ATP production.*

Reply: As suggested by the reviewer, we assessed mitochondrial respiration and ATP production in KRAS-mutant *versus* KRAS-wt cells by measuring oxygen consumption rate (OCR) and total ATP levels. Our results show that KRAS IEC-6 cells have higher OCR levels and produced relatively more ATP than Control cells (see figure below). Interestingly, we found that ATP7A depletion decreased total ATP levels in KRAS-mutant CRC cell lines (HCT116 and DLD-1) (Extended data Figure 7g), suggesting that ATP7A contributes to the loading of cuproenzymes involved in mitochondrial respiration. Consistent with this, we found that ATP7A depletion significantly reduced the activity of the Cu-dependent enzyme cytochrome c oxidase (CCO/COX) (Extended data Figure 7h), which is directly correlated to maximal mitochondrial respiration (PMID: 24218578).

4) *Macropinocytosis may mediate uptake of other several growth factors and cytokines. Therefore, it is important to show that in vivo anti-tumor effect of EIPA is dependent on the blockage of Cu intake but not other factors. What happens if treat Cu-deficient diet-fed xenograft mice with EIPA? Does anti-tumor effect of EIPA go away?*

Reply: We thank the reviewer for pointing out this experimental control, which was actually part of our initial experimental plan. Unfortunately, we found during preliminary experiments that the combination of Cu deficiency and EIPA treatment resulted in the loss of all animals subjected to this protocol. These mice rapidly reached critical endpoints due to weight loss, visible dehydration, and general signs of distress, which led to their prompt euthanasia. These endpoints were carefully chosen based on what is expected from prolonged Cu deficiency and EIPA treatment (PMID: 29959467; 9782174), and were approved by the Université de Montréal Animal Care Committee, which applies guidelines set by the Canadian Council on Animal Care (CCAC). Notwithstanding the likely opposition that we would face from the Animal Care Committee if we were to propose to repeat this control condition, we feel that it would be ethically questionable to do so.

That said, we do not feel that this control is required to conclude that macropinocytosis is involved in Cu uptake by KRAS-mutant cells *in vivo*. We have been very careful not to over interpret our findings, by stating that “...blocking macropinocytosis reduces Cu uptake, Cu-dependent signaling, as well as tumor growth”. In fact, we argue that many nutrients, in addition to Cu, are likely to enter KRAS-mutated cells via macropinocytosis. Therefore, we expect that EIPA treatment would also have Cu-independent effects on tumor growth, as was shown by its involvement in amino acid uptake (PMID: 23665962). The multiple roles of macropinocytosis in tumor growth make it difficult, if not impossible, to identify a single cause for the decrease in tumor growth observed in response to EIPA treatment. This is why we decided to focus our *in vivo* efforts on demonstrating that macropinocytosis is involved in Cu uptake (*via* CCS).

5) Does overexpression of ATP7A confer the resistance against Copper overload and also growth advantage in Kras-WT cells?

Reply: We thank the reviewer for this very interesting suggestion. Our plan was to test this by measuring the fitness of Control IEC-6 cells stably overexpressing ATP7A. Unfortunately, while we have obtained an ATP7A expressing construct from Dr. Van de Sluis (PMID: 21667063), he also warned us that the plasmid was very difficult to propagate in bacteria. This is in fact a well-documented problem that appears to be due to the instability of the ATP7A cDNA (PMID: 27337370). We tried different approaches to increase plasmid stability in bacteria, but could not successfully propagate plasmids to be used for subcloning ATP7A into retroviral vectors.

6) *Function of ATP7A may differ between intestinal cells and other cell types. Is this Copper dependence of Kras-mutated cells general finding even in other cancer types such as PDAC?*

Reply: Again, this is an interesting suggestion. The literature suggests that ATP7A plays a role in several types of cancer. However, we have not tested if the observed ATP7A dependence is specific to gastrointestinal cancers compared to other KRAS-driven cancers, such as lung and PDAC. The literature suggests that ATP7A is differentially regulated in intestinal cells compared to peripheral tissues, due to the special and early role played by ATP7A in systemic Cu

homeostasis (PMID: 28931909). Nevertheless, the role of ATP7A in pancreatic cancer is virtually unknown except for a proteomic study suggesting that it is uniquely expressed in PDAC compared to normal tissues (PMID: 16285947). While Cu chelation using TTM has promising anti-tumor effects on pancreatic cancer cell lines (PMID: 24218578), we believe that testing the function of ATP7A in PDAC should be considered as an exciting future direction.

7) Isn't it possible to directly measure the intracellular Cu levels instead of other surrogate markers such as CCS levels? It may convince us of your findings more.

Reply: We thank the reviewer for this comment, and we certainly agree with the need for alternative approaches to measure Cu levels in cells. As pointed out in our response to reviewer 3, cellular Cu can be divided into a canonical, static pool (i.e., Cu bound to enzymes such as cytochrome c oxidase) and an exchangeable, labile pool (i.e., Cu bound to chaperones such as CCS). The latter form of Cu is more bioavailable and can participate in dynamic cell signaling pathways (PMID: 25692243). As indicated by reviewer 2, CCS levels are a well-established readout that correlates with bioavailable Cu levels (PMID: 16950140; 28931909). Our results show that KRAS-mutated cells have decreased CCS levels compared to control counterparts (Figure 3c), suggesting that oncogenic KRAS increases Cu bioavailability in cells. We also show that inhibition of macropinocytosis increases CCS levels in KRAS-mutated cells, but not in control cells (Figure 5c), suggesting that macropinocytosis is an important entry route for Cu that becomes bioavailable in KRAS-mutated cells.

While there are alternative technologies to measure intracellular Cu levels, such as inductively coupled plasma-mass spectrometry (ICP-MS), these types of approaches detect total metal content, which does not necessarily reflect the biologically active or bioavailable Cu pools (PMID: 16950140; 22900086; 28617866; 28931909). For example, it was shown that CTR1 knockout leads to Cu accumulation and a paradoxical increase in CCS levels, suggesting that enhanced Cu levels do not necessarily correlate with its bioavailability and/or capacity to regulate the activity of cuproenzymes (PMID: 16950140). As an alternative to CCS levels, we decided to use Cu-sensing fluorescent probes, which provide relative information on the bioavailability and distribution of labile Cu pools in cells (PMID: 25692243). For this, we initiated a collaboration with Dr. Christoph Fahrni (Georgia Institute of Technology), who developed a ratiometric Cu(I)-selective fluorescent probe (crisp-17) that was recently used to probe labile Cu pools in ATP7A-mutated patient fibroblasts (PMID: 31160463). Using this probe, we observed that KRAS-mutant IEC-6 and CRC cells responded significantly higher to DTDP treatment, an oxidant that was shown to release thiol-bound intracellular Cu (Figure 3d,g). Notably, the kinetics for gradual regression of the initial ratio increase upon DTDP addition was significantly slower for KRAS cells relative to wild-type counterparts, suggesting an increased depth of the bioavailable thiol-bound Cu pool in KRAS mutant cells. We also observed that treatment of KRAS-mutated cells with inhibitors of macropinocytosis exhibited a small but significant decrease in fluorescence ratio, suggesting reduced bioavailable Cu levels (Figure 5d). Consistent with these results, the CCS levels were increased upon inhibition of macropinocytosis (Figure 5c). Finally, using ICP-MS we also observed that the levels of total Cu are higher in KRAS-mutated CRC compared to wild-type counterparts (Extended data Figure 5I).

Altogether, the additional ratiometric imaging and ICP-MS data demonstrate that Cu bioavailability is increased in oncogenic KRAS, and that macropinocytosis is an important entry route, thus reflecting our main conclusions.

8) Regarding CRISPR library method, there are several questions and concerns.

A) It is undescribed how the gene-level differential essentiality scores were calculated. Is it based on the average change of all designed gRNAs targeting same gene?

Reply: Bayes factors (BF) were calculated using the BAGEL algorithm, which is a confidence measure and considers the fold-change of all gRNAs targeting the same gene (PMID: 27083490). A positive BF signifies that the gene affects cell fitness and negative BF signifies that the gene is dispensable to cell fitness. Differential essentiality scores were calculated after subtracting the BF of KRAS from Control, and presented as Log₂ FC. If Log₂ FC ≥ 2, genes were considered differentially essential to KRAS cells. The methods section has been edited to better explain the calculations.

B) Number of predefined reference genes (essential and non-essential) used are fewer than the original BAGEL method. Only 193 genes, which may potentially cause bias. Assumption is Kras mutation status does not alter gene essentiality regarding these genes.

However, if it's not the case and kras mutation significantly alter dependency on many essential genes or independency on many non-essential genes, it might not be appropriate to directly compare BF calculated from BAGEL algorithms between Kras-mutated cells and Kras-wt cells. In addition, recent paper use the modified version of BAGEL to remove confident cancer genes from the reference set (Behan FM *et al.*, Nature 2019). Please consider this point.

Reply: The reference sets of essential and non-essential genes were originally compiled on the basis of over 400 shRNA screens and have been used extensively in scoring hundreds of CRISPR screens - many of which are KRAS mutant. Behan *et al.*, for instance found that most BAGEL essential genes were pan-cancer essential genes in a panel of 197 cell lines. Also, as shown in Figure 2c, while there are a small number of reference essential genes that had a shift in essentiality between Control and KRAS (i.e., *Myc*), the assumption that the mutation does not alter dependency appears to be valid for most of these genes. In addition, Figure 2b shows that Control and KRAS cells roughly follow the same distribution, so we are confident that it would not cause a substantial bias. We also performed a second screen analysis using a different tool, *MaGeck*, and confirmed that ATP7A scored as a significant hit for negative selection in the KRAS mutant line (FDR=0.017).

C) It is generally difficult to keep the library complexity in *in vivo* setting, which makes difficult us to perform negative CRISPR screening *in vivo*. Please show read count distribution in cells and tumors. How many gRNAs indeed remained inside tumors? Are most of gRNAs targeting non-essential genes still remained?

Reply: Our results suggest that the coverage is as expected in the *in vivo* screen. The library we used in this study is small and coverage is even better than what can sometimes be seen in other screens (>30 reads reads with 75% of the library).

D) Contents of surfaceome-based gRNA library was not clearly written. Although it was written as gRNAs against cell-surface protein-coding genes differentially modulated by KRAS in the method, how are these genes selected? Gene list in the library seems not to match genes identified in their LC/MS-based surface protein assay.

Reply: We hope to provide clarifications with respect to the reviewer's concerns. We have edited our manuscript to more clearly indicate how target genes were selected for generating the gRNA library. The manuscript now indicates that "Genes coding for cell-surface proteins were selected based on their significant up- or down-regulation ($\text{Log}_2 \text{FC} \geq +2$ or -2) at the surface of KRAS-mutated IEC-6 cells compared to wild-type counterparts. Rat gRNAs against selected cell-surface protein-coding genes were pooled with gRNAs against previously described reference genes that are known to be essential or non-essential to cell survival. This library, collectively referred to as KRAS-library (3,732 gRNAs in total), was synthesized as 58-mer oligonucleotides and amplified by PCR in a pooled format".

9) It is unclear how authors chose 20 genes to show the correlation of Kras-specific changes between IEC-6 and CRC specimens in Figure 1C. It seems to be arbitrary selected. More clear explanation or comparison of global transcriptome change should be done.

Reply: We apologize for the lack of clarity on how we designed the heatmap. Using cBioPortal (<https://www.cbioportal.org/>), we first isolated the list of genes most differentially expressed in KRAS-altered compared to KRAS-unaltered CRC specimens (TCGA, 594 samples). We then compared this list to the genes differentially expressed in KRAS-mutated versus Control IEC-6 cells, and derived common genes between the two datasets. We also filtered the dataset to identify genes that were previously reported to be regulated by oncogenic KRAS. We then selected 20 genes (10 over- and 10 under-expressed) to generate the heatmap shown in Figure 1c.

Our objective from that analysis was to isolate several mutant KRAS-selective genes that are significantly altered in CRC specimens and confirmed by the literature. We are comparing KRAS-specific changes occurring in the transcriptome of a rat cell lines *versus* human CRC specimens. As this comparison is limited between two different organisms and two different types of datasets (*in vitro* against *in vivo*), we intentionally focused our analysis to isolate several mutant KRAS-selective genes among the most significant according to the literature. We have accordingly edited our conclusions in the main body of the manuscript.

10) Please perform gene annotation enrichment analysis using TCGA CRC data with or w/o Kras mutation to assess whether similar enrichment as intestinal cells is seen.

Reply: As requested, we performed this analysis but as expected and as mentioned above, gene annotation enrichments were not similar between both datasets due to the limitations of this comparison.

11) Description of Supplementary Table 2 is unclear. Table contains Fold-change (Kras/Control) but there is no raw number of peptide for each group. Moreover, fold changes of some genes are incomputable but \log_2 fold change show 15. This arbitrarily value transformation needs to be explained in the methods, if applied.

Reply: As suggested by the reviewer, we have added peptide counts despite the fact that our quantifications are based on peptide intensities. Regarding the second part of the question, there are indeed some proteins identified only in the KRAS or Control condition, as pointed out by the reviewer, making it impossible to calculate fold-change values. Therefore, drawing inspiration from other studies, including Hörmann et al., (PMID: 26699813), we arbitrarily assigned +15 if proteins were only present in the KRAS condition, and -15 if only identified in Control condition. The value “±15” was chosen as it was higher than our highest and lowest KRAS/Control fold-change, respectively. As suggested by the reviewer, we properly edited the methods section by including these explanations.

12) It is unclear what N/A means in Extended data Figure 2F.

Reply: As mentioned above, we could not calculate some of the fold-change values for some of the identified proteins, as they were unique to Control or KRAS cells. We acknowledge the lack of clarity and we have modified the legend for Extended data Figure 2F.

***** end of point-by-point response to reviewers' comments *****

REVIEWERS' COMMENTS:

Reviewer #1 (Remarks to the Author):

1. As previously suggested by reviewer #1, the authors have considerably modified their manuscript to highlight its strengths. For example, highlighting the advantages of taking alternative chemoproteomic approaches and the complementarity of joint transcriptomics and cell-surface proteomics induced through oncogenic KRAS. This nicely adds value to the manuscript and highlights their approach.

2. Regarding protein inference stringency details, the authors have modified the methods section significantly (e.g., indicating FDR for peptide and protein $<0.5\%$, minimum 2 peptides per inferred protein, considerations of oxidative modification of cysteine/ methionine and serine, threonine or tyrosine phosphorylation and proteins being selected after detection in at least two biological replicates and minimum of 2 unique peptides verified using the neXtProt checker). These represent major improvements and considerably elevate the stringency of the data. The community's ongoing reliance (including some influential proteomics leaders) on reporting shorter peptides of 6, 7 or 8 amino acids is common and currently not optimal (unless 2 longer peptides are co-observed). Overall, this reviewer is now satisfied by the many efforts to increase the protein inference stringency at which they have used their data.

3. The authors have modified the manuscript to emphasize complementarity of chemoproteomics approaches and how the CSB and CSC employ different biotinylating reagents and their explanation of possible differences in the accessibility of functional groups and/or labelling of particular types of cell-surface proteins. I agree that this is a strength of the study, and the authors have emphasized this nicely in the manuscript.

4. Their explanation as to why they chose label-free approaches over TMT after recognition of reduced peptide yield for MS/MS is clear, as is their choice to optimize surfaceome methods using label-free quantitative proteomics.

5. Their explanation of choice of IEC-6 cells and several CRC cell lines in the revised manuscript is now far more than adequate.

6. The authors have also now adequately discussed why they did not perform transcriptomic or proteomic screens using Cu-supplement diet or chelators, in regards to off-target effects due to Cu supplementation and the possibility that chelators themselves could possibly have off-target effects by blocking other important metals.

Reviewer #2 (Remarks to the Author):

The revised manuscript generally improves on the original submission by offering a more thorough explanation of the relevant screening methodologies used, highlighting methodological and conceptual strengths, providing more thorough analyses of intracellular copper levels in various experiments, and providing more exhaustive testing of the Ras-specific effects of targeting ATP7A in colorectal cancer cells. The authors have made substantial efforts to address most of the major concerns raised in the review. However, as outlined below, certain conclusions need be tempered in accordance with the relative strength of the new data presented. On the whole, however, the revised version appears to be suitable for publication in Nature Communications.

Specific comments:

1. The authors offer new data to test the "differential essentiality" of ATP7A in KRAS WT versus KRAS mutant cells, both in an isogenic setting and in a more physiological context using multiple human colorectal cancer cell lines. These data appear in Figure 2. In Figure 2e, the isogenic IEC-6 cell lines are tested. Here treatment with siRNAs targeting ATP7A actually increase the cell number of control cells relative to the nontargeting siRNA. However, treatment of KRAS mutant cells with the ATP7A siRNA does decrease cell number by about one-third. This is promising and does suggest that KRAS sensitizes cells to ATP7A knockdown, but it is likely that these effects are a function of decreased cell proliferation rather than frank cell death (given the size of the effect and

depending on the doubling time of the IEC cells). One would argue that this is not characteristic of an essential gene, although it is noted that the knockdown leaves a substantial amount of ATP7A protein, which likely limits the effects seen in these experiments. In light of these caveats, it would be prudent to note the limitations of this experiment in the text. The same sorts of caveats apply to figure 2f, in which the human CRC lines are tested. Some small effects on cell number are seen in the WT KRAS line (CACO-2), though not as substantial as those seen in the KRAS mutant lines (HCT116, SW620). This difference is likely a function of cell doubling time however. (CACO-2, 32-80 hours; HCT116, 17-36 hours; SW620, 20-26 hours—according to ExpASY bioinformatics resource portal), but this is not tested. Nonetheless, the KRAS mutant cells do appear more sensitive, but the argument that ATP7A is a synthetic lethal target in these experiments (as opposed to the CRISPR screen) are difficult to make. The authors should rephrase their conclusion to suggest that these data are consistent with the differential essentiality defined for ATP7A in the CRISPR screen rather than confirmatory.

2. The new measures of intracellular labile Cu are very convincing. One minor comment is that the y-axis range for the data presented in Figure 5d (0.25-0.45) match those used in Figure 3d or 3g (0-2, 0-4, respectively). In Figure 5d, the y-axis for fluorescence ratio using the crisp-17 reagent does not go to 0. Clearly the data are statistically significant, but the axis range inflates the degree of difference, which should simply be acknowledged as being relatively small compared to what was seen in other experiments, likely because macropinocytosis is not the exclusive mode of Cu import into these cells.

3. The thorough response with regards to the EIPA control arm on the in vivo study and do understand the ethical concerns regarding the mouse work given the deleterious effects of combined EIPA treatment and Cu-deficiency are well appreciated. It is recommended that the authors provide this information to the readers in a concise way in the main body of the text, since it will be a naturally questioned omission. The statement “We find that KRAS-mutated cells acquire copper via a CTR1-independent mechanism involving macropinocytosis, which appears to be required for growth,” is too strong, since (1) it is still difficult to claim that macropinocytosis changes Cu levels directly in vivo, since only surrogates of Cu levels are measured for the xenograft experiment and (2) tumors still grow in all conditions.

Reviewer #3 (Remarks to the Author):

In the revised manuscript by Aubert et al., the authors provide additional correlative evidence that increased ATP7A levels are associated with KRAS mutation status, an increased labile Cu pool, and macropinocytosis. These additional data support the original findings and invoke new questions based on recent publications by Chung et al. PNAS and Liao et al. Nat Comm. The former demonstrates that labile Cu pools are reduced in response to oncogenic transformation by KRAS, which is in contrast to the data presented here. The latter demonstrates via a different approach that Cu plays an integral role in colon cancer tumorigenesis by regulating a XIAP signaling axis that would in turn reduce CCS protein levels as suggested by Aubert et al. Prior to acceptance, the authors should address the similarities and differences in their findings with these new manuscripts and address the CTR1 independence of Cu uptake to more adequately test the connection between Cu and macropinocytosis.

Major Points

1. The authors indicate that this occurs in a CTR1-independent fashion without directly testing it. A. The authors should perform both ICP-MS and imagine for Cu levels in the context of CTR1 knockdown/knockout in the presence or absence of EIPA to ascertain the contribution of macropinocytosis to Cu uptake.

Reviewer #4 (Remarks to the Author):

Authors mostly responded to my concerns and questions in appropriate manner, and revised the manuscript accordingly. Their novel findings are scientifically supported by sufficient experimental evidences in the revised manuscript and thus become more convincing. Additional experiments overexpressing ATP7A would further strengthen the conclusion, but nevertheless, I believe that this paper would greatly advance our further understanding of ras-driven cancer and provide potential therapeutic option toward this deadly cancer. I don't have any further comment on this paper.

Takahiro Kodama

REVIEWERS' COMMENTS:

Reviewer #1 (Remarks to the Author):

1. As previously suggested by reviewer #1, the authors have considerably modified their manuscript to highlight its strengths. For example, highlighting the advantages of taking alternative chemoproteomic approaches and the complementarity of joint transcriptomics and cell-surface proteomics induced through oncogenic KRAS. This nicely adds value to the manuscript and highlights their approach.

2. Regarding protein inference stringency details, the authors have modified the methods section significantly (e.g., indicating FDR for peptide and protein <0.5%, minimum 2 peptides per inferred protein, considerations of oxidative modification of cysteine/ methionine and serine, threonine or tyrosine phosphorylation and proteins being selected after detection in at least two biological replicates and minimum of 2 unique peptides verified using the neXtProt checker). These represent are major improvements and considerably elevate the stringency of the data. The community's ongoing reliance (including some influential proteomics leaders) on reporting shorter peptides of 6, 7 or 8 amino acids is common and currently not optimal (unless 2 longer peptides are co-observed). Overall, this reviewer is now satisfied by the many efforts to increase the protein inference stringency at which they have used their data.

3. The authors have modified the manuscript to emphasize complementarity of chemoproteomics approaches and how the CSB and CSC employ different biotinylating reagents and their explanation of possible differences in the accessibility of functional groups and/or labelling of particular types of cell-surface proteins. I agree that this is a strength of the study, and the authors have emphasized this nicely in the manuscript.

4. Their explanation as to why they chose label-free approaches over TMT after recognition of reduced peptide yield for MS/MS is clear, as is their choice to optimize surfaceome methods using label-free quantitative proteomics.

5. Their explanation of choice of IEC-6 cells and several CRC cell lines in the revised manuscript is now far more than adequate.

6. The authors have also now adequately discussed why they did not perform transcriptomic or proteomic screens using Cu-supplement diet or chelators, in regards to off-target effects due to Cu supplementation and the possibility that chelators themselves could possibly have off-target effects by blocking other important metals.

Reply: We are thankful that we were able to satisfy reviewer #1 during the revision process for all the comments from 1-6 and thank him/her for the comments that improved the quality of our manuscript.

Reviewer #2 (Remarks to the Author):

The revised manuscript generally improves on the original submission by offering a more thorough explanation of the relevant screening methodologies used, highlighting methodological and conceptual strengths, providing more thorough analyses of intracellular copper levels in various experiments, and providing more exhaustive testing of the Ras-specific effects of targeting ATP7A in colorectal cancer cells. The authors have made substantial efforts to address most of the major concerns raised in the review.

However, as outlined below, certain conclusions need be tempered in accordance with the relative strength of the new data presented. On the whole, however, the revised version appears to be suitable for publication in Nature Communications.

Reply: We are thankful that we were able to satisfy reviewer #2 in the revision process and thank him/her for the comments that helped us improve the quality of our manuscript.

Specific comments:

1. The authors offer new data to test the “differential essentiality” of ATP7A in KRAS WT versus KRAS mutant cells, both in an isogenic setting and in a more physiological context using multiple human colorectal cancer cell lines. These data appear in Figure 2. In Figure 2e, the isogenic IEC-6 cell lines are tested. Here treatment with siRNAs targeting ATP7A actually increase the cell number of control cells relative to the nontargeting siRNA. However, treatment of KRAS mutant cells with the ATP7A siRNA does decrease cell number by about one-third. This is promising and does suggest that KRAS sensitizes cells to ATP7A knockdown, but it is likely that these effects are a function of decreased cell proliferation rather than frank cell death (given the size of the effect and depending on the doubling time of the IEC cells). One would argue that this is not characteristic of an essential gene, although it is noted that the knockdown leaves a substantial amount of ATP7A protein, which likely limits the effects seen in these experiments. In light of these caveats, it would be prudent to note the limitations of this experiment in the text. The same sorts of caveats apply to figure 2f, in which the human CRC lines are tested. Some small effects on cell number are seen in the WT KRAS line (CACO-2), though not as substantial as those seen in the KRAS mutant lines (HCT116, SW620). This difference is likely a function of cell doubling time however. (CACO-2, 32-80 hours; HCT116, 17-36 hours; SW620, 20-26 hours—according to ExPASY bioinformatics resource portal), but this is not tested. Nonetheless, the KRAS mutant cells do appear more sensitive, but the argument that ATP7A is a synthetic lethal target in these experiments (as opposed to the CRISPR screen) are difficult to make. The authors should rephrase their conclusion to suggest that these data are consistent with the differential essentiality defined for ATP7A in the CRISPR screen rather than confirmatory.

Reply: We thank the reviewer for this detailed comment. The text has been edited for both CRC and IEC-6 cells' counting data with ATP7Ai in figure 2, to state that the results are “consistent with the differential essentiality of ATP7A as suggested by the CRISPR screens” and the “confirmatory” phrase has been removed.

2. The new measures of intracellular labile Cu are very convincing. One minor comment is that the y-axis range for the data presented in Figure 5d (0.25-0.45) match those used in Figure 3d or 3g (0-2, 0-4, respectively). In Figure 5d, the y-axis for fluorescence ratio using the crisp-17 reagent does not go to 0. Clearly the data are statistically significant, but the axis range inflates the degree of difference, which should simply be acknowledged as being relatively small compare to what was seen in other experiments, likely because macropinocytosis is not the exclusive mode of Cu import into these cells.

Reply: We thank the reviewer for this constructive criticism. We have edited the graph presentation in Figure 5d to include scale from (0-0.5) instead of (0.25-0.45). We also changed the text in figure 5 to conclude the results as “Furthermore, we found find that inhibition of macropinocytosis exhibited a small but statistically significant decrease in the basal fluorescence ratio of crisp-17 in KRAS cells (Fig. 5d), suggesting that macropinocytosis contributes to the supply of bioavailable Cu”. We have also edited the discussion to state that “We expect that macropinocytosis may not be the exclusive mode of Cu supply in

these cells and other mechanisms such as autophagy or other low-affinity transporters could also play a role (PMID: 27339113)".

3. The thorough response with regards to the EIPA control arm on the in vivo study and do understand the ethical concerns regarding the mouse work given the deleterious effects of combined EIPA treatment and Cu-deficiency are well appreciated. It is recommended that the authors provide this information to the readers in a concise way in the main body of the text, since it will be a naturally questioned omission. The statement "We find that KRAS-mutated cells acquire copper via a CTR1-independent mechanism involving macropinocytosis, which appears to be required for growth," is too strong, since (1) it is still difficult to claim that macropinocytosis changes Cu levels directly in vivo, since only surrogates of Cu levels are measured for the xenograft experiment and (2) tumors still grow in all conditions.

Reply: We thank the reviewer for these comments. We have taken this into consideration and edited the manuscript to mention "alternative or non-canonical Cu supply route" and removed the term "CTR1-independent mechanism". To inform the reader, we have also indicated the toxicity we observed with the combined EIPA treatment and Cu-deficiency. We hope that these changes satisfy reviewer #2.

Reviewer #3 (Remarks to the Author):

In the revised manuscript by Aubert et al., the authors provide additional correlative evidence that increased ATP7A levels are associated with KRAS mutation status, an increased labile Cu pool, and macropinocytosis. These additional data support the original findings and invoke new questions based on recent publications by Chung et al. PNAS and Liao et al. Nat Comm. The former demonstrates that labile Cu pools are reduced in response to oncogenic transformation by KRAS, which is in contrast to the data presented here. The latter demonstrates via a different approach that Cu plays an integral role in colon cancer tumorigenesis by regulating a XIAP signaling axis that would in turn reduce CCS protein levels as suggested by Aubert et al. Prior to acceptance, the authors should address the similarities and differences in their findings with these new manuscripts and address the CTR1 independence of Cu uptake to more adequately test the connection between Cu and macropinocytosis.

Reply: We are thankful for the comments that helped us improve the quality of our manuscript. We appreciate the reviewer for bringing to our attention new publications that involve KRAS transformation and/or copper levels in colorectal cancers. With regards to Liao et al., Nat Comm (PMID: 32060280) the authors showed that copper levels were increased in colorectal cancer organoids and KRAS-mutated Ls174t cells via IL-17-STEAP4-XIAP dependent mechanism. In our cells, STEAP4 was transcriptionally downregulated in KRAS cells compared to Control IEC-6. While we do not rule out that other mechanisms could alter Cu supply, we have not directly tested the effect of inflammation on Cu levels in KRAS cells. Notably, it was promising to see that TTM reduced metastatic burden of CRC in this study.

With regards to the Chung et al., PNAS (PMID: 31451653), Mouse embryonic fibroblasts (MEF) cells expressing conditional oncogene KRAS^{G12D} had lower fluorescence ratios using copper sensors, and they concluded that loosely-bound labile Cu(I) pools were reduced upon oncogenic transformation. Consistent with this data the basal fluorescence ratios (which represent loosely bound and labile Cu(I) pools) were not increased in KRAS cells, but addition of DTDP a thiol-inducing oxidant to increase Cu levels from Cu-buffering chaperones significantly increased Cu levels in KRAS (Figure 3d). This suggests that increased Cu is more efficiently buffered in KRAS IEC-6 cells. We also note that the cell line used is different from intestinal epithelial cells, and we expect that ATP7A and KRAS transformation have different effects in different organ tissues (PMID: 28931909).

Major Points

1. The authors indicate that this occurs in a CTR1-independent fashion without directly testing it.

A. The authors should perform both ICP-MS and imagine for Cu levels in the context of CTR1 knockdown/knockout in the presence or absence of EIPA to ascertain the contribution of macropinocytosis to Cu uptake.

Reply: We have taken the reviewer's comments into consideration and edited the manuscript to remove the "CTR1-independent" statement to instead use "alternative or non-canonical Cu supply route". We hope that this satisfies reviewer #3 as an alternative for the major point/experiment raised.

Reviewer #4 (Remarks to the Author):

Authors mostly responded to my concerns and questions in appropriate manner, and revised the manuscript accordingly. Their novel findings are scientifically supported by sufficient experimental evidences in the revised manuscript and thus become more convincing. Additional experiments overexpressing ATP7A would further strengthen the conclusion, but nevertheless, I believe that this paper would greatly advance our further understanding of ras-driven cancer and provide potential therapeutic option toward this deadly cancer. I don't have any further comment on this paper.

Reply: We are happy that we have provided sufficient information in the revision process to satisfy reviewer #4. We thank him for his help to improve our manuscript.